# Learning to Counter: Stochastic Feature-based Learning for Diverse Counterfactual Explanations

## Abstract

Interpretable machine learning seeks to understand the reasoning process of complex black-box systems that are long notorious for lack of explainability. One growing interpreting approach is through counterfactual explanations, which go beyond why a system arrives at a certain decision to further provide suggestions on what a user can do to alter the outcome. A counterfactual example must be able to counter the original prediction from the black-box classifier, while also satisfying various constraints for practical applications. These constraints exist at trade-offs between one and another presenting radical challenges to existing works. To this end, we propose a stochastic learning-based framework that effectively balances the counterfactual trade-offs. The framework consists of a generation and a feature selection module with complementary roles: the former aims to model the distribution of valid counterfactuals whereas the latter serves to enforce additional constraints in a way that allows for differentiable training and amortized optimization. We demonstrate the effectiveness of our method in generating actionable and plausible counterfactuals that are more diverse than the existing methods and particularly in a more efficient manner than counterparts of the same capacity.

## 1 Introduction

Recent advances in machine learning, especially the successes of deep neural networks, have promoted the use of these systems in various real-world applications. Such models provide remarkable predictive performance yet often at a cost of transparency and interpretability. This has sparked controversy over whether to rely on algorithmic predictions for critical decision making, from graduate admission (Waters & Miikkulainen, 2014; Acharya et al., 2019), job recruitment (Ajunwa et al., 2016) to high-stakes cases of credit assessment (Lessmann et al., 2015) or criminal justice (Lipton, 2018; Gifford, 2018). Progress in interpretable machine learning offers interesting solutions to explaining the predictive mechanisms of black-box models. One useful interpreting approach is through counterfactual examples, which sheds light on what modifications to be made to an individual's profile that can counter an unfavorable decision outcome from a black-box classifier. Such explanations explore what-if scenarios that suggest possible recourses for future improvement. Counterfactual explanability indeed has important social implications at both personal and organizational level. For instance, job applicants who get rejected by the CV screening algorithm of a company are likely to benefit from feedbacks like 'getting 1 more referral' or 'being fluent in at least 2 languages', which would help them better prepare for future applications. At organizational level, by engaging with job candidates in this way as a form of advocating for transparency in decision making, companies can improve employer branding and attractiveness to top talents. Internally, organizations can also validate whether any prejudice or unfairness towards a particular group is implicitly introduced in historical data and consequentially embedded in the classifiers producing biased decisions.

**Related works.** Recent years have seen an explosion in literature on counterfactual explanability, from works that initially focused on one or two specific characteristics or families of models to those that can deal with multiple constraints and various model types. There have been many attempts to summarize major themes of research and discuss open challenges in great depth. We therefore refer readers to Karimi et al. (2020b); Verma et al. (2020); Guidotti (2022) for excellent surveys of

methods in this area. We here focus on reviewing algorithms that can support multiple or even *diverse* counterfactual generations - a property that has received less attention.

Dealing with the combinatorial nature of the task, earlier works commonly adopt mixed integer programming (Russell, 2019), genetic algorithms (Sharma et al., 2020), or SMT solvers (Karimi et al., 2020a). Another recent popular approach is gradient-based optimization (Mothilal et al., 2020; Bui et al., 2022). In a similar fashion with adversarial learning (Goodfellow et al., 2014), it involves iteratively perturbing the input data point according to an objective function that incorporates desired constraints. The whole idea of *diversity* is to explore different combinations of features and feature values that can counter the original prediction while accommodating various user needs. To support *diversity*, Russell (2019) in particular enforces hard constraints on the current generations to be different from the previous ones. Such a constraint will however be removed whenever the solver cannot be satisfied. Meanwhile, Mothilal et al. (2020) and Bui et al. (2022) add another loss term for *diversity* using Determinantal Point Processes (Kulesza et al., 2012), whereas the other works only demonstrate the capacity to generate multiple counterfactuals via empirical results. Moreover, all of these algorithms are computationally expensive. Along the line, Redelmeier et al. (2021) attempts to model the conditional likelihood of mutable features given the immutable features using the training data. They then adopt Monte Carlo sampling to generate counterfactuals from this distribution and filter out samples that do not meet counterfactual constraints. Amortized optimization emerges as a more effective strategy that explicitly models the distribution of counterfactual examples via a generative model such as a Variational auto-encoder (VAE) (Mahajan et al., 2019; Pawelczyk et al., 2020; Downs et al., 2020) or a Markov decision process (MDP) model under a reinforcement learning setting (Verma et al., 2022). Obtaining such a distribution, sampling of counterfactuals can therefore be done straightforwardly.

**Contributions.** In this paper, we propose a learning-based framework diverging markedly from the previous approaches. We reformulate the combinatorial search task into a stochastic optimization problem that can be solved efficiently via gradient descent. Whereas the previous works model the generative distributions via MDP Verma et al. (2022), VAE or VAE-based counterparts (Mahajan et al., 2019; Pawelczyk et al., 2020; Downs et al., 2020), we construct a learnable generation module $\mathcal{G}$ that directly models the conditional distributions of individual features such that they form a valid counterfactual distribution when combined.

Another point of difference of our framework lies in the usage of Bernoulli sampling to ensure that only minimal changes are introduced to the generative counterfactuals. In prior works, standard metrics such as L1 or L2 is often used to penalize the distance between the counterfactual and original data point. Verma et al. (2020) criticizes this approach as non-obvious, especially for handling categorical features. Avoiding the use of distance measures, we optimize a feature selection module $\mathcal{S}$ to output a Bernoulli distribution for each feature representing the likelihood of the feature being mutated. $\mathcal{S}$ is a flexible module that can adapt to different user-defined constraints about mutability of features.

Similar to most works, our framework is developed to deal with heterogeneous tabular data. However, instead of one-hot encoding every categorical feature and treat each level as an individual numerical feature, we propose the opposite strategy: to discretize the numerical features and treat them as categorical. The benefits are four-fold: (1) we can conveniently apply one functional form over feature distributions, which then only requires one reparameterization trick feature-wise; (2) it helps expand the original input space that may later support generalization; (3) we believe that it yields more useful explanations and easier for human users to follow the suggestions than forcing them to meet hard requirements from specific numerical values; (4) it helps reduces privacy risks when revealing the counterfactual suggestions to the public. To facilitate end-to-end differentiable training, we employ the Gumbel-Softmax reparameterization trick for effective treatment of categorical features. This is the first time this strategy is used in this line of research.

Our contributions can be summarized as follows

- We introduce **L**earning to **C**ounter (**L2C**) - a stochastic feature-based learning approach for generating counterfactual explanations that address the counterfactual desirable properties in a single end-to-end differentiable framework.

- We employ amortized optimization to **efficiently** generate **actionable** and **diverse** counterfactual explanations at $100\%$ **validity** and adequate trade-off with **sparsity**. Our method additionally enforces **plausibility** by optimizing the generative examples on the input **data manifold** while **maintaining one-hotness** in the output representations for the categorical features.
- Through extensive experiments on real-world datasets, L2C is shown to balance the trade-offs among counterfactual constraints more effectively than the existing methods whereas achieving more diverse explanations and sufficient robustness in out-of-distribution settings. To the best of our knowledge, we are the first amortized method that supports diverse local counterfactual generations.

In the following, we provide a detailed description of the characteristics highlighted in bold.

## 2 DESIDERATA OF COUNTERFACTUAL EXPLANATIONS

The ultimate goal of this line of research is to provide practical guidelines as to what actions an individual can take to achieve a desired outcome. Desiderata of counterfactual explanations have been extensively discussed in previous literature (Karimi et al., 2020b; Verma et al., 2020; Guidotti, 2022; Verma et al., 2022). For pragmatic reasons, this paper focuses on counterfactual explanations with the following characteristics:

- *Validity:* By definition, a counterfactual example must flip the original black-box outcome.
- *Actionability:* Counterfactual explanations should be specific to individual user preferences and only suggest actionable or feasible changes. In particular, changes should be made on *mutable* features e.g., Work Experience or SAT scores, while leaving *immutable* features unchanged e.g., Gender or Ethnicity.
- *Sparsity:* Counterfactuals should be close to the original example where a minimal number of features are modified.
- *Diversity:* Diverse explanations are preferable to capture different preferences from the same user so that they can freely explore multiple options to select the best fit.
- *Plausibility*: Plausible or realistic counterfactuals are to lie close to the training data manifold and obey the input domain. An example of implausible instances is one with a numerical feature such as Age being above 100 years old, or a categorical feature with more than one category is assigned non-zero values in its one-hot representation.
- *Scalability:* Inference should be done simultaneously and efficiently for multiple input examples.

While *validity* and *actionability* are the two must-have criteria to generate a practical counterfactual explanation, satisfying some, if not all, of the other constraints at the same time is a challenging task. First, there is a trade-off between *validity* and *sparsity* (Verma et al., 2020). To counter the original outcome, it is naturally easier to change a large number of features without considering whether the features are mutable or not. On the other hand, strongly enforcing *sparsity* results in a smaller subset of features that can be changed, which can compromise *diversity* since we expect counterfactual states to differ from one to another substantially.

With respect to *Plausibility*, this property is often violated when categorical features are not handled properly (Verma et al., 2020). A common approach to pre-process categorical features is through one-hot encoding. However, the output representation of a counterfactual example may not maintain one-hotness or probabilistic format where the sum of all levels equals to $1$. This can lead to a discrepancy where its equivalent one-hot representation (i.e., the plausible representation) fails to produce the counterfactual outcome. Appendix B.2 will investigate this issue in more detail. Lastly, *scalability* is an important consideration and particularly significant in the face of *diversity*. This is because the majority of previous methods optimize each input data point separately, and individual runs are additionally required to produce several counterfactuals. These algorithms are thus highly time-expensive for large datasets. Monte Carlo sampling (Redelmeier et al., 2021) and amortized optimization (Mahajan et al., 2019; Downs et al., 2020; Verma et al., 2022) are recently adopted to boost efficiency. However, none of these methods addresses the *diversity* constraint nor is equipped with appropriate treatment for categorical features. We now explain how our framework L2C satisfies all the above constraints.

# 3 STOCHASTIC FEATURE-BASED COUNTERFACTUAL LEARNING

## 3.1 PROBLEM SETUP

Let $\mathcal{X}$ denote the input space where $\boldsymbol{x}$ is an input vector with $N$ features of both continuous and categorical types. As discussed previously, we discretize the numerical features into equal-sized buckets, which gives us an input of $N$ categorical features wherein each feature $x_i$ has $c_i$ levels. We apply one-hot encoding on each feature and flatten them into a single input vector $\boldsymbol{z} \in \{0, 1\}^D$ where $D = \sum_{i=1}^{N} c_i$. Concretely, feature $x_i$ is now represented by the vector $\boldsymbol{z}_i \in \mathbb{O}_{c_i}$ where the set of one-hot vectors $\mathbb{O}_{c_i}$ is defined as $\{0, 1\}^{c_i} : \sum_{j=1}^{c_i} z_{ij} = 1$.

Let $f$ be the black-box classifying function and $y = f(\boldsymbol{x})$ be the decision outcome on the input $\boldsymbol{x}$. A valid counterfactual example $\widetilde{\boldsymbol{x}}$ associated with $\boldsymbol{x}$ is one that alters the original outcome $y$ into a desired outcome $y' \neq y$ with $y' = f(\widetilde{\boldsymbol{x}})$. Let $\widetilde{\boldsymbol{z}}$ denote the corresponding one-hot representation of $\widetilde{\boldsymbol{x}}$.

Actionability indicates that some features can be *mutable* (i.e., changeable), while others should be kept *immutable* (i.e., unchangeable). Without loss of generality, let us impose an ordering on the set of $N$ features such that the first $K$ features are mutable features (i.e., the ones that can be modified) and denote $\mathbb{K} := \{1, ..., K\} \subset \{1, ..., N\}$. For each mutable feature (i.e., $\boldsymbol{x}_i$ or the one-hot vector $\boldsymbol{z}_i$ with $i \in \mathbb{K}$), we aim to learn a local feature-based perturbation distribution $P(\widetilde{\boldsymbol{z}}_i \mid \boldsymbol{z})$ where $\widetilde{\boldsymbol{z}}_i \in \mathbb{O}_{c_i}$, while leaving the immutable features unchanged.

To assure a fair comparison, we train the same black-box classifier for both ours and the baseline methods. Specifically, the input to the classifier is the representation where only categorical features are one-hot encoded while numerical features are retained at their original values. Later, whenever necessary to consult the black-box model, for every numerical feature $i$, we convert the one-hot representation $\widetilde{\boldsymbol{z}}_i$ back to $[a_i, b_i]$ where $[a_i, b_i]$ is the original value range of the feature.

## 3.2 METHODOLOGY

We now detail how L2C works and addresses each counterfactual constraint. Our framework is summarized in Figure 1.

For each mutable feature $\boldsymbol{z}_i$ with $i \in \mathbb{K}$, we learn a **local feature-based perturbation distribution** $P(\widetilde{\boldsymbol{z}}_i \mid \boldsymbol{z})$ (i.e., $\tilde{z}_i \in \mathbb{O}_{c_i}$), which is a categorical distribution $\text{Cat}(\boldsymbol{p}_i \mid \boldsymbol{z})$ with category probability $\boldsymbol{p}_i = [p_{i1}, p_{i2}, ..., p_{ic_i}]$. We form a counterfactual example $\widetilde{\boldsymbol{z}}$ by concatenating $\widetilde{\boldsymbol{z}}_i \sim \text{Cat}(\boldsymbol{p}_i \mid \boldsymbol{z})$ for the mutable features and $\boldsymbol{z}_i$ for the immutable features. To achieve *validity*, we learn the local feature-based perturbation distribution by maximizing the chance that the counterfactual examples $\widetilde{\boldsymbol{z}}$ counter the original outcome on $\boldsymbol{x}$.

Additionally, learning local feature-based perturbation distributions over the mutable features allows us to conduct a global counterfactual distribution $P(\widetilde{\boldsymbol{z}} \mid \boldsymbol{z})$ over the counterfactual examples $\widetilde{\boldsymbol{z}}$ defined above. Sampling from this distribution naturally leads to multiple counterfactual generations efficiently, and we also expect that individual samples $\widetilde{\boldsymbol{z}}_i$ together can form diverse combinations of features, thereby promoting *diversity* within the generative examples.

As previously discussed, too much of *diversity* can compromise *sparsity*. Dealing with this constraint, for each mutable feature $\boldsymbol{z}_i$, we propose to learn a **local feature-based selection distribution** that generates a random binary variable $\boldsymbol{s}_i \sim \text{Bernoulli}(\pi_i \mid \boldsymbol{z})$ wherein we replace $\boldsymbol{z}_i$ by $\widetilde{\boldsymbol{z}}_i \sim \text{Cat}(\boldsymbol{p}_i \mid \boldsymbol{z})$ if $\boldsymbol{s}_i = 1$ and leave $\tilde{\boldsymbol{z}}_i = \boldsymbol{z}_i$ if $\boldsymbol{s}_i = 0$. Therefore, the formula to update $\widetilde{\boldsymbol{z}}_i$ is

$$\widetilde{\boldsymbol{z}}_i = (1 - \boldsymbol{s}_i)\boldsymbol{z}_i + \boldsymbol{s}_i\widetilde{\boldsymbol{z}}_i.$$

The benefit of having $\boldsymbol{\pi} = [\pi_i]_{i \in \mathbb{K}}$ is thus to control *sparsity* by adding one more channel to decide if we should modify a mutable feature $\boldsymbol{z}_i$. Appendix B.4 presents an ablation study showing that without the selection distribution, the perturbation distribution alone can generate diverse counterfactuals but it changes plenty of mutable features. Meanwhile, optimizing the selection distribution jointly helps harmonize the trade-off between *diversity* and *sparsity*.

## 3.3 OPTIMIZATION OBJECTIVE

In this section, we explain how to design the building blocks of our framework L2C. As shown in Figure 1, our framework consists of two modules: a **counterfactual generator** $\mathcal{G}$ and a **feature selector** $\mathcal{S}$. The **counterfactual generator** $\mathcal{G}$ is used to model the feature-based perturbation distribution, while **feature selector** $\mathcal{S}$ is employed to model the feature-based selection distribution.

Specifically, given a one-hot vector representation $z$ of a data example $x$, we feed $z$ to $\mathcal{G}$ to form $\mathcal{G}(z) = [\mathcal{G}_i(z)]_{i \in \mathbb{K}}$. We then apply the softmax activation function to $\mathcal{G}_i(z)$ to define the feature-based local distribution (i.e., $\mathrm{Cat}(p_i \mid z)$)) for $z_i$ as

$$p_{ij}(z) = \frac{\exp\{\mathcal{G}_{ij}(z)\}}{\sum_{k=1}^{c_i} \exp\{\mathcal{G}_{ik}(z)\}}, \forall j = 1, ..., c_i.$$

$\mathcal{S}$ takes $z$ to form $\mathcal{S}(z) = [\mathcal{S}_i(z)]_{i \in \mathbb{K}}$. We then apply the Sigmoid function to $\mathcal{S}_i(z)$ to define the feature-based selection distribution (i.e., $\mathrm{Bernoulli}(\pi_i \mid z)$) for $z_i$ as

$$\pi_i(z) = \frac{1}{1 + \exp\{-\mathcal{S}_i(z)\}}.$$

To encourage *sparsity* by reducing the number of mutable features chosen to be modified, we regularize $\mathcal{S}$ through L1-norm $\|\boldsymbol{\pi}(z)\|_1$ with $\boldsymbol{\pi}(z) = [\pi_i(z)]_{i \in \mathbb{K}}$.

To summarize, given an one-hot vector representation $z$ of a data example $x$, we use $\mathcal{G}$ to work out the local feature-based perturbation distribution $\mathrm{Cat}(p_i(z))$ for every $i \in \mathbb{K}$. We then sample $\widetilde{z}_i \sim \mathrm{Cat}(p_i(z))$ for every $i \in \mathbb{K}$. Subsequently, we use $\mathcal{S}$ to work out the local feature-based selection distribution $\mathrm{Bernoulli}(\pi_i(z))$ for every $i \in \mathbb{K}$. We then sample $s_i \sim \mathrm{Bernoulli}(\pi_i \mid z)$ and update $\widetilde{z}_i = (1 - s_i)z_i + s_i\widetilde{z}_i$ for every $i \in \mathbb{K}$. Finally, we concatenate $\widetilde{z}_i$ for $i \in \mathbb{K}$ and $z_i$ for $i \notin \mathbb{K}$ to form the counterfactual example $\widetilde{z}$.

$\mathcal{G}$ and $\mathcal{S}$ are parameterized with neural networks over total parameters $\theta$. For $\widetilde{z}$ to be a *valid* and *sparse* counterfactual associated with a desired outcome $y'$, we propose the following criterion

$$\min_\theta \left[ \mathbb{E}_{\widetilde{z}}\left[\mathrm{CE}(f(\widetilde{z}), y')\right] + \alpha\, \mathbb{E}_z\left[\|\boldsymbol{\pi}(z)\|_1\right] \right], \tag{1}$$

where $f$ is the black-box function, CE is the cross-entropy loss, $\|\cdot\|_1$ is L1-norm, $\alpha$ is a loss weight.

Notice that $\widetilde{z}$ formed by concatenating many one-hot vectors is an incompatible representation to the classifier $f$, which in fact requires both continuous and one-hot features. In our implementation, we reconstruct the numerical features by taking the middle point of the range corresponding to the selected level. We refer to this process as **one-hot decoding**. Specifically, the input to the one-hot decoder is $\widetilde{z} = [\widetilde{z}_i]_{i=1}^N$. If the feature $i$ originally is a categorical feature, we set $\widetilde{x}_i = \widetilde{z}_i$. Otherwise, we set $\widetilde{x}_i = a_i + \frac{(2k-1)(b_i - a_i)}{2}$, which is the middle point of the interval $[a_i + (k-1)(b_i - a_i), a_i + k(b_i - a_i)]$ where $[a_i, b_i]$ is the range of the feature $i$ and $\widetilde{z}_i$ corresponds to the level $k \in \{1, ..., c_i\}$ (i.e., $\widetilde{z}_{ik} = 1$ and $\widetilde{z}_{ij} = 0$ if $j \neq k$). To facilitate the continuous relaxation in Section 3.4, we rewrite

$$\widetilde{x}_i = \sum_{j=1}^{c_i} \widetilde{z}_{ij}\left[a_i + \frac{(2j-1)(b_i - a_i)}{2}\right]. \tag{2}$$

The final optimization objective is now given as

$$\min_\theta \left[ \mathbb{E}_{\widetilde{x}}\left[\mathrm{CE}(f(\widetilde{x}), y')\right] + \alpha\, \mathbb{E}_z\left[\|\boldsymbol{\pi}(z)\|_1\right] \right]. \tag{3}$$

## 3.4 REPARAMETERIZATION FOR CONTINUOUS OPTIMIZATION

Our L2C involves multiple sampling rounds back and forth to optimize the networks. To make the process continuous and differential for training, we adopt the following reparameterization tricks:

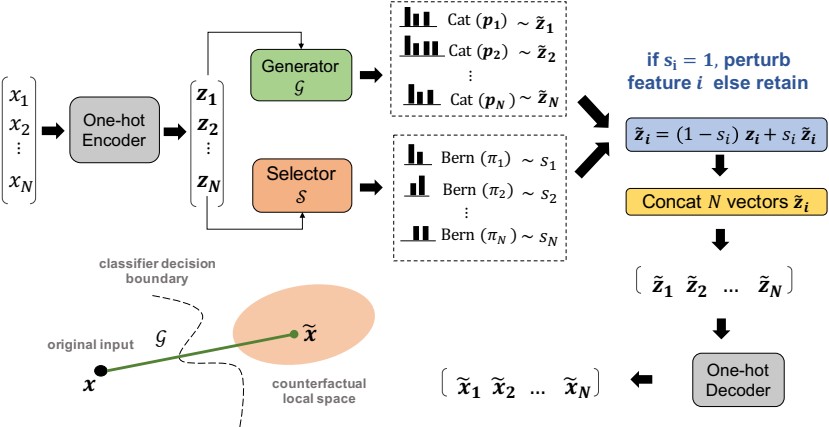

Figure 1: For illustration purpose only, the figure assumes all features are mutable. Given an input $\boldsymbol{x}$, we discretize the numerical features into categorical levels and one-hot encode all features into representations $\boldsymbol{z}$. For every feature $i$, the generator $\mathcal{G}$ learns a local perturbation distribution $\text{Cat}(\boldsymbol{p}_i|\boldsymbol{z})$ so that together they form a distribution of *diverse* counterfactual representations $\widetilde{\boldsymbol{z}}$. Intuitively, $\mathcal{G}$ aims to construct a "bridge" across the decision boundary travelling from the input to a local space of counterfactuals. Simultaneously, the selector $\mathcal{S}$ learns to output the distribution Multi-Bernoulli$(\boldsymbol{\pi}|\boldsymbol{z})$ capturing the probability of each feature $i$ being modified. Every pair of feature sample $(\widetilde{\boldsymbol{z}}_i, s_i)$ is passed through operation in the blue box, which decides whether to accept the change being made to the feature $i$ given by $\widetilde{\boldsymbol{z}}_i$ and enforces feature *actionability* accordingly. The output is then decoded into the representations $\widetilde{\boldsymbol{x}}$ compatible with the black-box system. $\mathcal{G}$ and $\mathcal{S}$ are jointly trained via back-propagation according to the objective given in Eq. (3) where the first loss term ensures $\mathcal{G}$ produces *valid* counterfactuals whereas the second term encourages $\mathcal{S}$ to produce *sparse* feature combinations. We assure *plausibility* by optimizing both distributions on input data manifold and utilizing reparameterization trick to handle categorical features. Finally, through amortized optimization, L2C supports *scalability* for efficient generations of multiple counterfactuals.

**1) Sampling $\widetilde{z}_i \sim \mathbf{Cat}(\boldsymbol{p}_i \mid \boldsymbol{z})$** : To obtain differentiable counterfactual samples, we adopt the classic temperature-dependent Gumbel-Softmax trick (Jang et al., 2016; Maddison et al., 2016). Given the categorical variable $\mathbf{z}_i$ with category probability $\left[p_{i1}, p_{i2}, ..., p_{ic_i}\right]$. The relaxed representation is sampled from the Categorical Concrete distribution as $\widetilde{z}_i \sim \text{Cat-Concrete}(\log p_{i1}, ..., \log p_{ic_i})$ by

$$\widetilde{z}_{ij} = \frac{\exp\left\{(\log p_{ij}(\boldsymbol{z}) + G_j)/\tau\right\}}{\sum_{k=1}^{c_i} \exp\left\{(\log p_{ik}(\boldsymbol{z}) + G_k)/\tau\right\}}.$$

with temperature $\tau$, random noises $G_j$ independently drawn from Gumbel distribution $G_t = -\log(-\log u_t)$, $u_t \sim \text{Uniform}(0, 1)$. As discussed, we apply this mechanism consistently to the one-hot representations of all features. The continuous relaxation of Eq. (2) can be gained by simply using the one-hot relaxation $\widetilde{z}_i$.

**2) Sampling $s_i \sim \mathbf{Bernoulli}(\pi_i \mid \boldsymbol{z})$** : We again apply the Gumbel-Softmax trick to relax Bernoulli variables of 2 categories. With temperature $\tau$, random noises $G_{i0}$ and $G_{i1} \sim G_t = -\log(-\log u_t)$, $u_t \sim \text{Uniform}(0, 1)$, the continuous representation $\boldsymbol{s}_i$ is sampled from Binary Concrete distribution as $\boldsymbol{s}_i \sim \text{Bin-Concrete}(\pi_i, 1 - \pi_i)$ by

$$s_i = \frac{\exp\{\left(\log \pi_i(\boldsymbol{z}) + G_{i1}\right)/\tau\}}{\exp\{\left(\log(1 - \pi_i(\boldsymbol{z})) + G_{i0}\right)\}/\tau\} + \exp\{\left(\log \pi_i(\boldsymbol{z}) + G_{i1}/\tau\right)\}}.$$

## 4 EXPERIMENTS

We experiment with 4 popular real-word datasets: German Credit (Dua & Graff, 2017), Graduate Admission (Acharya et al., 2019), Student Performance (Cortez & Silva, 2008) and Small Business Administration (SBA) (Li et al., 2018). For the last 2 datasets, we adopt the setup from Bui et al. (2022), which introduces temporal and geospatial shifts respectively to the testing sets. The goal

is to additionally validate how robust the proposed method is to covariate shifts. For each dataset, we select a fixed subset of immutable features based on our domain knowledge and suggestions from (Verma et al., 2022). While implementing the black-box classifiers and the baseline methods, we standardize numerical features to unit variance and one-hot encode categorical features. Note again that, for our method only, we discretize numerical features into equal-sized buckets and decode the numerical features back to their original representations whenever feeding them to the black-box model. Appendix A describes our tasks and model design in greater detail with the discretization setup specifically reported in Appendix A.3. Our code repository can be accessed at `https://anonymous.4open.science/r/L2C-AD51/`.

Table 1: Description of quantitative evaluation metrics. $\mathbb{C}$ denotes a set of counterfactual samples generated by an interpreting method for a given input example.

| Property | Metric | Description |
|---|---|---|
| Validity | `Validity` `Coverage` | - Proportion of samples in $\mathbb{C}$ can counter the original black-box decision outcome.
- `Coverage` $= 100\%$ if there exists at least 1 valid counterfactual in $\mathbb{C}$. |
| Sparsity / Actionability | `Sparsity` `Prox Num` `Prox Cat` | - Proportion of features being changed, averaged over the number of samples in $\mathbb{C}$.
- L1 distance of the input data point to a counterfactual sample across the numerical features, averaged over the number of samples in $\mathbb{C}$.
- Hamming distance of the input data point to a counterfactual sample across categorical features, averaged over the number of samples in $\mathbb{C}$. |
| Diversity | `Diversity` | - Hamming distance of a pair of counterfactual samples across all features where numerical features are discretized. The metric is averaged over all pairs of samples in $\mathbb{C}$. |
| Plausibility | `Manifold Dist.` `Valid Cat` | - L1 distance of a counterfactual sample to 5 closest training data points across the numerical features. Nearest instances are determined by a pre-trained kNN algorithm. The metric is averaged over the number of samples in $\mathbb{C}$.
- Proportion of categorical features in a counterfactual sample having a proper one-hot or probabilistic representation in which sum of the values at all levels equal 1. The metric is averaged over the number of samples in $\mathbb{C}$. |

### 4.1 PERFORMANCE METRICS

Following the past works Mothilal et al. (2020); Redelmeier et al. (2021); Verma et al. (2022), Table 1 summarizes the commonly used metrics for quantitatively assessing how well a method satisfies the constraints outlined in section 2. As for `Diversity`, a widely adopted measure is the pairwise distance between counterfactual examples, with distance defined separately for numerical and categorical features (Mothilal et al., 2020; Redelmeier et al., 2021). Though this approach is meaningful for interpreting categorical features, we however find it quite obscure for numerical features. This motivates us to discretize numerical features again when computing `Diversity`, which captures how often a feature gets altered as well as how much the change is - specifically via how often it switches to a different categorical level. The computation of `Diversity` only considers valid counterfactuals, so if none of the examples in $\mathbb{C}$ are valid, `Diversity` is set to zero.

### 4.2 BASELINES AND EVALUATION SETUP

We compare our method against existing amortized approaches: Feasible-VAE (Mahajan et al., 2019), CRUDS (Downs et al., 2020), FastAR (Verma et al., 2022), and popular non-amortized methods that support multiple counterfactual generation: DiCE(Mothilal et al., 2020), CERTIFAI (Sharma et al., 2020), MCCE (Redelmeier et al., 2021) and COPA (Bui et al., 2022). DiCE offers several search strategies: Random, KD-tree, or Genetic algorithm. DiCE-Random is adopted in our experiment since it is reported to consistently yield the best performance across datasets (Verma et al., 2022). We do not consider MOC (Dandl et al., 2020), which does not support Python implementation, or MACE (Karimi et al., 2020a) due to its convergence negatively depending on the SAT solver, which is also extremely expensive on large datasets (Verma et al., 2022).

Here we consider a general setting of binary classification where a counterfactual outcome $y'$ is opposite to the original outcome $y$, whether $y$ is positive or negative. From each method, we generate a set of 100 counterfactual explanations. During generation, most methods, including ours, require multiple iterations of searching for the optimal set of counterfactuals based on the optimization constraints. To assure a fair comparison on efficiency, a global maximum time budget of 5 minutes is imposed to search for a set of 100 counterfactuals per input sample. For non-amortized baselines,

the algorithms are run directly on the testing sets, while for amortized methods, we train the base generative models on the training sets and use the testing sets only for evaluation. We tune the generative models via grid search and report the best settings in Appendix C, which also includes the description about our special treatment for FastAR.

## 5 RESULTS AND DISCUSSION

We train each black-box classifier on 5 model initializations and report the average results in Table 2 and Table 4. We also report `Time` that records the total inference time in seconds for a set of 100 counterfactuals of all testing inputs. Note that COPA has only been shown to work effectively on linear classifiers. Appendix D provides several illustrative examples for qualitative assessment.

Given the same time budget, our method L2C succeeds in generating $100\%$ valid counterfactuals with full coverage. Together with DiCE and Feasible-VAE, L2C first satisfies the most important criterion of a counterfactual explanation and resolves the trade-off against *validity*. Since the trade-off between *sparsity* and *diversity* often hinders comparison, we attempt to approximate the numerical proximity level of the closest baseline DiCE by setting a sparsity threshold between 0 and 1 during inference. The thresholds are chosen in such a way that no more than 2 numerical features get modified in one sample. Valid counterfactuals exceeding the threshold are filtered out. Recall that we have specified a fixed set of immutable features for each dataset, based on which we can work out the maximum sparsity threshold a counterfactual explanation should adhere to (See Appendix A.1). Feasible-VAE, MCCE and CRUDS cannot fulfill this constraint for all datasets. This is one critical drawback of Feasible-VAE whose implementation in fact does not consider mutability of features. It is also evident that L2C is the second-fastest generator while consistently achieving the best *diversity* of all methods.

Table 2: Comparison of interpreting methods on in-distribution datasets and the corresponding black-box architectures. ↓ Lower is better. ↑ Higher is better. Bold indicates the best performance for each dataset. Underline indicates the best performance among methods achieving $100\%$ Validity.

| **Method** | Prox Num↓ | Prox Cat(%)↓ | Diversity (%)↑ | Sparsity (%)↓ | Validity (%)↑ | Coverage (%)↑ | Manifold Dist.↓ | Valid Cat (%)↑ | Time(s)↓ |
|---|---|---|---|---|---|---|---|---|---|
| German Credit (Logistic Regression) - Max. Sparsity: 80.00% | | | | | | | | | |
| **L2C (Ours)** | 0.44 | 30.01 | **37.81** | 29.46 | **100.00** | **100.00** | 0.44 | **100.00** | 33 |
| DiCE | 0.80 | 9.41 | 15.75 | 11.77 | **100.00** | **100.00** | 0.79 | 84.36 | 1,150 |
| F-VAE | 0.70 | 45.96 | 2.13 | 54.07 | **100.00** | **100.00** | 0.14 | 0.00 | 36 |
| COPA | 0.69 | 31.91 | 19.77 | 42.12 | 44.00 | 44.00 | 0.29 | 0.01 | 17,583 |
| MCCE | 1.05 | 66.17 | 33.40 | 71.24 | 48.74 | **100.00** | 0.35 | 57.44 | **2** |
| CERTIFAI | 1.16 | 51.77 | 9.92 | 59.00 | 53.52 | 61.49 | 0.48 | **100.00** | 50,258 |
| FastAR | **0.03** | **0.38** | 0.01 | **4.07** | 95.79 | 95.79 | 0.23 | 93.85 | 10,605 |
| CRUDS | 0.74 | 61.76 | 1.55 | 67.50 | 44.00 | 44.00 | 0.23 | 9.74 | 42,920 |
| Graduate Admission (Neural Network) - Max. Sparsity: 85.71% | | | | | | | | | |
| **L2C (Ours)** | 1.12 | 33.52 | **39.59** | 43.56 | **100.00** | **100.00** | 1.04 | **100.00** | 16 |
| DiCE | 1.17 | 22.00 | 33.06 | 33.75 | **100.00** | **100.00** | 1.30 | 73.50 | 412 |
| F-VAE | 0.86 | 86.50 | 1.54 | 92.29 | **100.00** | **100.00** | 0.19 | 0.00 | 16 |
| MCCE | 0.99 | 69.57 | 22.98 | 82.61 | 43.79 | 84.60 | 0.35 | 68.64 | **1** |
| CERTIFAI | 1.36 | **0.00** | 16.61 | 42.86 | 90.20 | 90.20 | 0.90 | **100.00** | 237 |
| FastAR | **0.66** | 58.69 | 1.07 | 72.03 | 87.41 | 87.41 | 0.32 | 72.30 | 5,405 |
| CRUDS | 0.84 | 75.75 | 1.39 | 86.14 | 51.48 | 52.00 | 0.20 | 11.00 | 21,460 |

As shown in Figure 2, L2C and DiCE are the two most effective in balancing the trade-offs among counterfactual constraints, in which L2C converges to this level of performance significantly faster. To quantify the effectiveness in balancing the trade-off between *sparsity* and *diversity*, we propose taking `Harmonic mean` of `Diversity` and `(1 - Sparsity)`. It follows the motivation of F1-score in measuring Precision against Recall, which thus equals to $\frac{2 \cdot \text{Diversity} \cdot (1 - \text{Sparsity})}{\text{Diversity} + (1 - \text{Sparsity})}$. Figure 3 shows the superiority of our method in addressing this trade-off based on the proposed score where L2C specifically outperforms the second-best method by $2.43\% - 18.33\%$ across datasets.

In Appendix B.3, we further evaluate the effects of different threshold levels, showing that setting the thresholds higher than the current levels can further improve inference time and *diversity* while maintaining a good balance against *sparsity*. We also refer readers to Appendix B.1 and B.2 where we discuss *plausibility* and *robustness* on out-of-distribution datasets in detail.

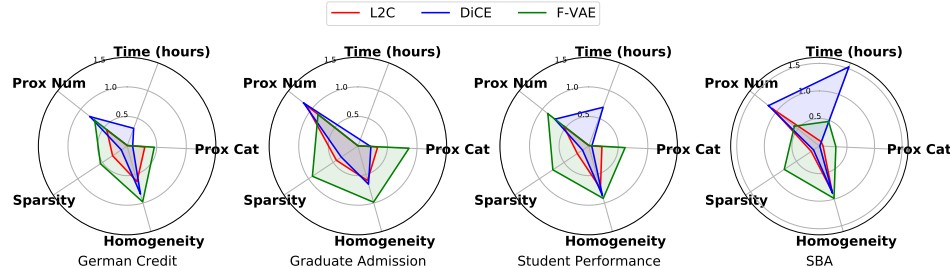

Figure 2: Visualizing methods with $100\%$ Validity: L2C, DiCE and Feasible-VAE across important constraints. `Homogeneity = 1 - Diversity` and `Time` is converted to hours for scaling down the value range. All metrics are preferably lower to be better. L2C lies closely towards the middle indicating that our method reaches a more effective balancing point.

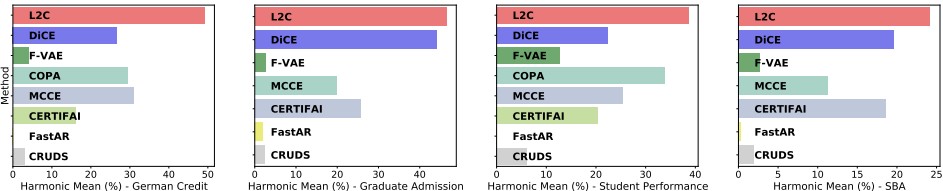

Figure 3: Comparing method capacity in balancing the trade-off of *diversity* and *sparsity* based on the `Harmonic mean` of `Diversity` and `(1 - Sparsity)`.

## 6 CONCLUSION AND FUTURE WORK

Diverging from the previous approaches, we contribute a novel counterfactual explanation framework that effectively resolves the trade-offs among counterfactual constraints. Here we target a broad class of differentiable machine learning classifiers. To fit non-differentiable models in our framework, one could use policy gradient (Sutton et al., 1999) or attempt to approximate such models as decision trees or random forests with a differentiable version (Yang et al., 2018; Lucic et al., 2022).

Although we have covered the essential desiderata of a counterfactual explanation, addressing causal relationships among features is beyond our scope in this paper. Handling correlational or even causal constraints is indeed a non-trivial challenge, which often requires domain knowledge and/or the true structural causal models. There are only a few works addressing this, notably Mahajan et al. (2019); Karimi et al. (2021); Verma et al. (2022), by assuming access to partial knowledge or utilizing constraints from user specifications. However, there have been no proper quantitative metrics to evaluate whether such a constraint is satisfied in practice. Recall that in this work, we assume the conditional distributions of each feature $P(x_i|x)$ to be independent. We thus can model the joint distribution among any $k$ features as $P(x_1, x_2, ..., x_k|x) = \prod_{i=1}^{k} P(x_i|x)$. Sharing the same motivation, one can extend L2C to impose constraints on such a joint distribution among the related features to ensure the generative samples respect causal constraints.

In recent years, there is an increasing concern regarding the privacy risks of counterfactual explanations (Shokri et al., 2021; Goethals et al., 2022). Handling the quality of explanations while taking security and privacy into account remains an interesting open challenging in this research area. Notice one of the main differences of our work is that we discretize data into categorical features. Thus, we believe this approach would help reduce the privacy risks of model stealing or sensitive data exposure when releasing the counterfactual suggestion to the public. Our future works will focus on tackling this issue more rigorously.

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

# A EXPERIMENTAL DETAILS

## A.1 DATASET STATISTICS

This section provides additional details about each dataset and experimental setup.

- **German Credit** (Dua & Graff, 2017): This dataset includes information of customers taking credit at a bank. The task is to classify a customer as a good (label 1) or bad (label 0) credit risk. The immutable features include *Foreign worker, Number of liable people, Personal status, Purpose* (Verma et al., 2022).

- **Graduate Admission** (Acharya et al., 2019): The set contains data of Indian students' applications to Master's program. The original target variable is an ordinal variable on the scale of $[0-1]$ indicating the chance of a student being admitted, where 1 indicates the highest chance. We set a threshold of 0.7 and re-categorize students as either "having higher chance" ($\geq 0.70$-label 1) or "having lower chance" ($< 0.70$-label 0). The binary classification task is to determine if a student profile has a higher chance of being successful at their application. The immutable feature is chosen to be *University Rating*.

- **Student Performance** (Cortez & Silva, 2008): This dataset records the performance of students at two schools Gabriel Pereira and Mousinho da Silveira. The task is to predict whether a student achieves a final score above average (label 1) or not (label 0). The train and test splits contain data of students from these two schools separately (Bui et al., 2022). The set of immutable features contains *Mother's education, Father's education, Family education support, First period grade*.

- **Small Business Administration (SBA)** (Li et al., 2018): The dataset presents data on loan approvals of small businesses from 1989 to 2014. The train split contains records of the period 1989-2006, and the test split includes observations of the period 2006-2014. The task is to classify a business with a high (label 1) or low risk (label 0) of loan default (Bui et al., 2022). The immutable features consist of *From Urban or Rural, New or Existing business, Active loan in Recession or not*.

Regarding the underlying black-box models, we experiment with Logistic Regression for the linear classifier and Neural Network for the non-linear classifier. We train linear classifiers on German Credit and Student Performance, and non-linear classifiers on Graduate Admission (with 3 layers and 40-dimensional hidden units) and SBA (with 3 layers and 30-dimensional hidden units). For each task, we further sample 20% random observations of the training sets as validation sets and train 5 black-box models with the same architecture but with different initializations. Since we have specified a fixed subset of immutable features, we therefore can derive the maximum sparsity level an explanation method should obey as

$$1 - \frac{\text{No. immutable features}}{\text{Total no. features}}$$

Table 3: Dataset statistics and Black-box performance.

| Dataset | Train/Dev/Test | No. Features | Black-box Architecture | Test Accuracy | Max. Sparsity |
|---|---|---|---|---|---|
| German Credit | 640/160/200 | 20 | Logistic Regression | 67.00% | 80.00% |
| Graduate Admission | 320/80/100 | 7 | Neural Network | 90.60% | 85.71% |
| Student Performance | 339/84/226 | 14 | Logistic Regression | 94.69% | 71.43% |
| SBA | 755/188/1159 | 12 | Neural Network | 90.39% | 75.00% |

### A.2 MODEL DESIGN

We parameterize $\mathcal{G}$ and $\mathcal{S}$ each with a $2-$layer neural networks. Each layer consists of a 50-dimensional dense layer, followed by a ReLU activation. The final layer of $\mathcal{G}$ is another dense layer that outputs a logit vector of the same dimension as the input, representing the counterfactual distribution. $\mathcal{S}$ produces a probability vector, thus taking Sigmoid as the output layer. We set the sparsity loss coefficient $\alpha = 0.001$ and use the same architecture for all tasks. We train our model with Adam optimizer for 200 epochs, at $\tau = 0.2$ and learning rate of 0.0001.

### A.3 DISCRETIZATION MECHANISM

Numerical features are discretized based on quantile values using Python function `qcut` [1], which requires specifying the maximum number of buckets/levels and later adjusts it depending on the input data distribution. We set the maximum buckets to be 4 such that each bucket has at least 100 data points.

Choosing very few is likely to cause under-fitting since there are very few useful combinatorial patterns that can counter the original label. We need sufficient diversity for effective learning. However, too many buckets are undesirable since it can hurt generalization due to some following reasons : (1) each bucket would contain too little data and the chosen middle value (used to decode one-hot vector as described in section 3.3) may not represent the bucket well, and (2) the model has more combinations of features to explore, thus can converge to sub-optimal combinations that cannot generalize well on unseen test points. The impact of discretization also depends on the proportion of numerical features in the data compared with categorical features. It would be less severe if there are more categorical features to provide sufficiently useful patterns for generalization. In this regard, we therefore decide to split data into equal-sized buckets in the hope of balancing the trade-off.

Table 8 reports how the numerical features of each dataset are discretized. For illustration purpose only, we round the edge values to the nearest whole number.

## B ADDITIONAL EXPERIMENTS

### B.1 ROBUSTNESS

Table 4 shows that L2C can adequately interpret out-of-distribution examples. One useful application is when, for example, a student from a different school wishes to transfer to the school which provides the data to train the models. Our system can inform the student of whether their academic performance suffices or what they should further do to increase the chance of being admitted. To explain this capability, since our framework optimizes local counterfactual distributions for each feature, L2C can cover a sufficiently large space of outputs to support generalization. This strategy is believed to be more effective than those optimizing for a single output data point, around which one can imagine that the distribution, if ever there was one, would have an extremely sharp peak, therefore more likely to fail under severe shifts. Note that robustness analysis is irrelevant for the non-amortized methods which are directly optimized on the testing sets in our experiments.

### B.2 PLAUSIBILITY

Table 4 also demonstrate the capability of maintaining proper representations for categorical features throughout the generative process. In our framework, the one-hot encoded representations are relaxed to be continuous during training through the temperature-dependent Gumbel-Softmax trick. In principle, as the temperature $\tau$ approaches zero, the continuous representations get closer to the binary vectors i.e., discrete Multi-Bernoulli samples, and we thus can expect the continuous samples to approximate the behavior of the discrete samples. During inference, we revert to discrete sampling and maintain one-hot representations in the generative samples.

There are many methods with `Valid Cat` far under $100\%$ meaning that the output representations for some categorical features are neither one-hot nor a probability vector. We often infer an explanation by selecting the categorical level with the maximum vector value. The inferred category by

---

[1]`https://pandas.pydata.org/docs/reference/api/pandas.qcut.html`

Table 4: Comparison of interpreting methods on out-of-distribution datasets and the corresponding black-box architectures. ↓ Lower is better. ↑ Higher is better. Bold indicates the best performance for each dataset. Underline indicates the best performance among methods achieving 100% Validity.

| Method | Prox Num↓ | Prox Cat(%)↓ | Diversity (%)↑ | Sparsity (%)↓ | Validity (%)↑ | Coverage (%)↑ | Manifold Dist.↓ | Valid Cat (%)↑ | Time(s)↓ |
|---|---|---|---|---|---|---|---|---|---|
| | | | Student Performance (Logistic Regression) - Max. Sparsity: 71.43% | | | | | | |
| **L2C (Ours)** | 0.54 | 22.42 | **25.79** | 23.02 | **100.00** | **100.00** | 1.63 | **100.00** | 36 |
| DiCE | 0.73 | 5.36 | 12.78 | 12.40 | **100.00** | **100.00** | 1.83 | 91.36 | 2,518 |
| F-VAE | 0.88 | 61.69 | 8.23 | 72.63 | **100.00** | **100.00** | 0.57 | 0.00 | 36 |
| COPA | 0.79 | 29.37 | 25.56 | 49.55 | 67.26 | 67.26 | 0.64 | 0.02 | 18,774 |
| MCCE | 0.96 | 63.65 | 24.97 | 74.03 | 60.98 | 93.10 | 0.84 | 71.38 | **1** |
| CERTIFAI | 1.05 | **0.00** | 11.88 | 28.57 | 98.23 | 98.23 | 1.69 | **100.00** | 834 |
| FastAR | **0.04** | 2.34 | 0.01 | **8.14** | 97.71 | 97.71 | **0.51** | 88.48 | 16,370 |
| CRUDS | 0.96 | 66.32 | 3.50 | 75.94 | 59.89 | 60.09 | 0.84 | 0.00 | 39,571 |
| | | | SBA (Neural Network) - Max. Sparsity: 75.00% | | | | | | |
| **L2C (Ours)** | 1.09 | 6.75 | **14.20** | 17.67 | **100.00** | **100.00** | 6.22 | **100.00** | 310 |
| DiCE | 1.18 | 0.69 | 11.08 | **12.78** | **100.00** | **100.00** | 6.69 | 97.78 | 5,522 |
| F-VAE | 0.58 | 29.85 | 1.45 | 76.62 | 96.59 | 97.33 | 1.42 | 0.00 | 1,716 |
| MCCE | 0.88 | 44.63 | 8.11 | 81.54 | 33.21 | 69.28 | 3.33 | 25.00 | **3** |
| CERTIFAI | 4.97 | **0.00** | 12.87 | 66.67 | 91.23 | 91.23 | 25.04 | **100.00** | 1,833 |
| FastAR | **0.51** | 21.91 | 0.16 | 63.41 | 86.77 | 86.77 | **0.97** | 86.13 | 55,175 |
| CRUDS | 0.70 | 63.51 | 1.08 | 87.84 | 58.51 | 62.24 | 2.12 | 24.01 | 248,719 |

convention corresponds to an equivalent one-hot representation, which is the plausible input format fed to the black-box classifier. Hence, a discrepancy can occur in that the output representation yields the desired outcome whereas the equivalent one-hot one may not, which poses a critical reliability issue (Guidotti, 2022).

We now investigate the violation of *plausibility* due to the failure to maintain one-hotness in the categorical features of a generative example. When processing categorical features, previous strategies include imposing regularization (Verma et al., 2020), clamping each one-hot column to be a specific categorical value (Wachter et al., 2017; Downs et al., 2020), relying on genetic algorithms or SMT solvers for automatic treatment (Karimi et al., 2020a; Schleich et al., 2021), or simply filtering them out (Lucic et al., 2022). It is clear that these techniques are insufficient to ensure all levels of a categorical feature are effectively handled, resulting in plenty of methods failing to achieve 100% on `Valid Cat` reported in Section 5.

We hypothesize that the black-box outcome on the output representation of a categorical feature is sometimes not aligned with the outcome on its equivalent one-hot representation. Specifically for a categorical feature, some methods output a continuous representation like $[0.05, 0.01, 0.02, 0.1]$ that may be not be summed up to 1. However, when returning the qualitative explanation to users, we need to return the inferred categorical level, which is 4 in this example. This level is thus equivalent to the one-hot representation $[0, 0, 0, 1]$. Since the compatible representation of category features to the black-box classifiers is in one-hot format, there may be a mismatch in the prediction on these two different representations for the categorical feature - that is, the continuous one can counter the original outcome while its equivalent one-hot one may not.

Given an output representation, for example $[0.05, 0.01, 0.02, 0.1]$, we adopt the categorical level with the maximum value as the explanation i.e., 4 in this case. Given this level, we encode the feature back into the proper one-hot representation i.e., $[0, 0, 0, 1]$. We assess `Validity` and `Coverage` of generative examples given the new representations and study this behavior in DiCE (Mothilal et al., 2020) and Feasible VAE (Mahajan et al., 2019) - the two baseline methods performing best on these metrics. Table 5 reveals that on some datasets, the performance of these methods deteriorates when evaluated on the equivalent one-hot representations.

## B.3 EFFECT OF SPARSITY THRESHOLDS

Figure 4 illustrates that increasing the sparsity thresholds imposed on the numerical features boosts time efficiency and *diversity* without compromising too much of *sparsity*. Here we also want to highlight the flexibility of our framework in controlling the quality of counterfactual generations during inference. Users can now freely set additional constraints to filer out unsatisfactory examples based on their preferences without re-training or re-optimization like the previous methods.

Table 5: Analysis of Validity and Coverage of DiCE and Feasible-VAE given the original Output and its Reverted one-hot representation. Bold highlights severe drops in performance. ↑ Higher is better.

| **Method** | **Version** | German Credit | Graduate Admission | Student Performance | SBA |
|---|---|---|---|---|---|
| Validity (%)↑ | | | | | |
| DiCE | Output | 100.00 | 100.00 | 100.00 | 100.00 |
| | Reverted | **58.20** | 91.28 | 98.12 | 98.93 |
| F-VAE | Output | 100.00 | 100.00 | 100.00 | 96.59 |
| | Reverted | 100.0 0 | 99.95 | 99.33 | **57.85** |
| Coverage (%)↑ | | | | | |
| DiCE | Output | 100.00 | 100.00 | 100.00 | 100.00 |
| | Reverted | 99.90 | 100.00 | 100.00 | 100.00 |
| F-VAE | Output | 100.00 | 100.00 | 100.00 | 97.33 |
| | Reverted | 100.00 | 100.00 | 100.00 | **72.10** |

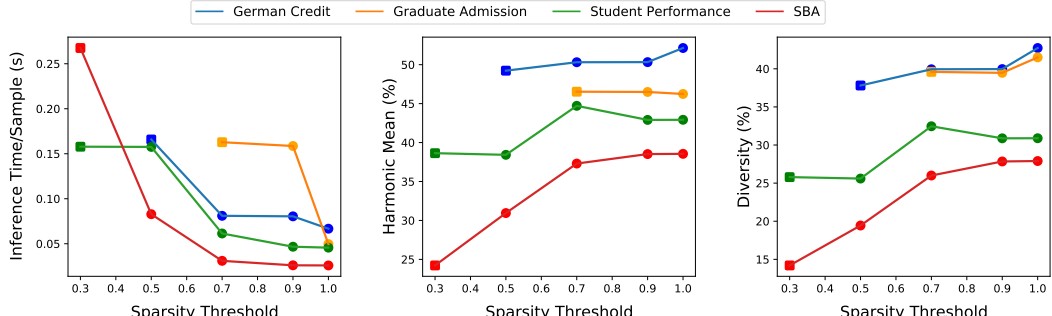

Figure 4: Effect of various sparsity thresholds on (**left**) time efficiency, measured via average inference time per sample (in seconds), (**middle**) the "balanced score" `Harmonic mean(Diversity, 1 - Sparsity)` and (**right**) diversity. The current levels reported in the main paper are denoted as square markers.

### B.4 ROLE OF THE FEATURE SELECTOR

Here we validate the importance of learning the local feature-based selection distribution via the feature selector $\mathcal{S}$. In this regard, we try removing $\mathcal{S}$ from L2C framework and replace the probability vector $\boldsymbol{\pi}(\boldsymbol{z})$ with a binary mask vector $\boldsymbol{m} \in [0, 1]^N$ where $m_i = 1$ if $i \in \mathbb{K}$ (i.e., a mutable feature) and $m_i = 0$ otherwise. We thus use $m_i$ in substitution of $s_i$ to update the counterfactual representations $\widetilde{z}_i$ as previously done. We now only optimize the generator $\mathcal{G}$ to learn the feature-based perturbation distribution, and the training objective Eq. (3) excludes the regularization term for *sparsity* accordingly. Table 6 investigates the performance of L2C under this alternative setup, in comparison with the proposed method that jointly optimizes $\mathcal{S}$ and $\mathcal{G}$. L2C now still achieves $100\%$ of `Validity` and `Coverage`, so we only report the relevant metrics that remain.

In this study, we do not impose any sparsity threshold because without the Selector $\mathcal{S}$, we lose the flexibility in tailoring the quality of counterfactual generations to potential user preferences. This is the first drawback. Second, though taking more time to infer explanations, the selector introduces significant *sparsity* to gain an effective balance of the trade-off against *diversity*. Furthermore, these results support our claim about the role of the generator $\mathcal{G}$ in that the perturbation distribution alone can yield impressively diverse explanations with *sparsity* remaining under the required maximum level. This is a benefit of learning an entire feature distribution in which sometimes the feature sample falls into the original input value i.e., $\widetilde{z}_i = z_i$ while combining adequately with other features.

Table 6: Analysis of L2C capacity to balance the trade-off between *diversity* and *sparsity* when optimized *without* the selector $\mathcal{S}$. ↓ Lower is better. ↑ Higher is better. Bold indicates the best performance for each dataset. `Time` records total inference time in seconds.

| Method | Prox Num↓ | Prox Cat(%)↓ | Diversity (%)↑ | Sparsity (%)↓ | Harmonic Mean (%)↑ | Time(s)↓ |
|---|---|---|---|---|---|---|
| German Credit (Logistic Regression) - Max. Sparsity: 80.00% | | | | | | |
| Without Selector | 1.41 | 55.84 | **52.66** | 62.47 | 43.83 | **5** |
| Proposed method | **0.82** | **29.56** | 42.73 | **33.13** | **52.14** | 13 |
| Graduate Admission (Neural Network) - Max. Sparsity: 85.71% | | | | | | |
| Without Selector | 1.54 | 58.67 | **48.71** | 76.38 | 31.81 | **1** |
| Proposed method | **1.25** | **31.85** | 41.51 | **47.79** | **46.25** | 5 |
| Student Performance (Logistic Regression) - Max. Sparsity: 71.43% | | | | | | |
| Without Selector | 1.37 | 46.32 | **41.03** | 54.52 | 43.14 | **4** |
| Proposed method | **1.14** | **32.93** | 40.97 | **43.43** | **47.52** | 8 |
| SBA (Neural Network) - Max. Sparsity: 75.00% | | | | | | |
| Without Selector | 2.89 | 12.74 | **36.79** | 70.91 | 32.49 | **15** |
| Proposed method | **1.77** | **6.26** | 27.90 | **37.64** | **38.55** | 30 |

## C  BASELINE EXPERIMENT

We tune the base generative models of amortized baselines under various different hyper-parameter settings. We determine the best settings via two metrics: `Coverage` and `Diversity`. When there is a trade-off, `Coverage` is chosen to be the deciding criterion.

For Feasible-VAE (Mahajan et al., 2019), we tune the hidden dimensions of the VAE encoder within $\{10, 30, 50, 70, 90\}$ and regularization term on Validity within $\{42, 62, 82, 102, 122\}$. For CRUDS (Downs et al., 2020), the base model is a Conditional Subspace Variational Auto-encoder (Klys et al., 2018). In the original paper, the network only has 1 hidden layer of 64 nodes, which we find to be of low capacity. We thus experiment with 2 layers and different hidden dimensions within $\{16, 32, 64\}$. For FastAR (Verma et al., 2022), the hyper-parameters include manifold distance $\lambda$ and entropy loss coefficient. Across datasets, Verma et al. (2022) shows the best `Coverage` under $\lambda = 0.1$. We thus set $\lambda = 0.1$ as well in our experiments, while focus on tuning the latter hyper-parameter within $\{0.01, 0.05, 0.1, 0.5, 1.0\}$. For the remaining hyper-parameters, we adopt the best values reported by the authors. Table 7 reports the best settings of these methods on each dataset.

**FastAR Specifics.**  A pre-trained FastAR model can only interpret one decision outcome chosen as the desired one (often the positive label). We must therefore train separate FastAR models on the positive and negative subsets and combine the results. We further find that although it is straightforward to obtain multiple generations in a single model, FastAR algorithm is optimized for one optimal counterfactual state for a given input. Thus in the hope of achieving better *diversity*, we train 100 different model initializations and accordingly collect a set of 100 explanations for evaluation. The inference time accumulates as a result, which is the reason why our reported results on time efficiency for FastAR are different from what are reported in the authors' paper.

Table 7: The best hyper-parameter settings for amortized methods.

| Method | Hyper-parameter | German Credit | Graduate Admission | Student Performance | SBA |
|---|---|---|---|---|---|
| F-VAE | Encoder hidden size | 10 | 30 | 10 | 70 |
| | Validity coefficient | 42 | 62 | 62 | 42 |
| CRUDS | Layer 1 hidden size | 16 | 16 | 64 | 16 |
| | Layer 2 hidden size | 16 | 16 | 16 | 16 |
| FastAR | Entropy coefficient | 0.01 | 0.01 | 0.1 | 1.0 |

Table 8: Discretization of numerical features in each dataset.

| Feature name | Bucket values | No. Data points |
|---|---|---|
| German Credit | | |
| Duration (Months) | ( 4 , 12 ] | 359 |
| | ( 12 , 24 ] | 411 |
| | ( 24 , 72 ] | 230 |
| Credit Amount | ( 248 , 1570 ] | 334 |
| | ( 1,570 , 3,410 ] | 333 |
| | ( 3,410 , 18,425 ] | 333 |
| Age | ( 19 , 28 ] | 334 |
| | ( 28 , 38 ] | 346 |
| | ( 38 , 75 ] | 320 |
| Graduate Admission | | |
| GRE score | ( 290 , 312 ] | 189 |
| | ( 312 , 323 ] | 148 |
| | ( 323 , 340 ] | 162 |
| TOEFL score | ( 92 , 104 ] | 176 |
| | ( 104 , 110 ] | 175 |
| | ( 110 , 120 ] | 149 |
| Undergraduate GPA | ( 7 , 8 ] | 170 |
| | ( 8 , 9 ] | 163 |
| | ( 9 , 10 ] | 167 |
| Student Performance | | |
| Age | ( 15 , 16 ] | 289 |
| | ( 16 , 17 ] | 179 |
| | ( 17 , 22 ] | 181 |
| School absences | ( -0 , 4 ] | 466 |
| | ( 4 , 32 ] | 183 |
| First period grade | ( -0 , 10 ] | 252 |
| | ( 10 , 13 ] | 245 |
| | ( 13 , 19 ] | 152 |
| Second period grade | ( -0 , 10 ] | 228 |
| | ( 10 , 11 ] | 103 |
| | ( 11 , 13 ] | 166 |
| | ( 13 , 19 ] | 152 |
| SBA | | |
| Loan term in months | ( -0 , 60 ] | 575 |
| | ( 60 , 84 ] | 671 |
| | ( 84 , 240 ] | 544 |
| | ( 240 , 306 ] | 312 |
| No. employees | ( -0 , 2 ] | 842 |
| | ( 2 , 3 ] | 221 |
| | ( 3 , 8 ] | 540 |
| | ( 8 , 650 ] | 499 |
| No. jobs created | ( -0 , 2 ] | 1658 |
| | ( 2 , 130 ] | 444 |
| No. jobs retained | ( -0 , 1 ] | 881 |
| | ( 1 , 4 ] | 619 |
| | ( 4 , 535 ] | 602 |
| Charged off amount | ( -107 , 15,074 ] | 1576 |
| | ( 15,074 , 1,509,538 ] | 526 |
| Gross amount approved | ( 4,260 , 30,026 ] | 588 |
| | ( 30,026 , 60,946 ] | 463 |
| | ( 60,946 , 300,056 ] | 537 |
| | ( 300,056 , 2,350,015 ] | 514 |
| SBA's guaranteed amount approved | ( 2,092 , 14,943 ] | 550 |
| | ( 14,943 , 41,541 ] | 501 |
| | ( 41,541 , 239,682 ] | 525 |
| | ( 239,682 , 2,115,002 ] | 526 |
| Proportion of gross amount guaranteed by SBA | ( 0.30 , 0.50 ] | 1052 |
| | ( 0.50 , 0.75 ] | 458 |
| | ( 0.75 , 1.00 ] | 592 |

# D  QUALITATIVE EXAMPLES

Here we illustrate some examples of our generated counterfactuals for each dataset. As discussed, we report the discretized values for numerical features, since we believe it provides more useful and flexible suggestions to human users. For illustration purpose only, we round the edge values of the numerical intervals to the nearest whole number. Immutable features are italicized.

Table 9: Counterfactual examples from German Credit dataset. *DM: Deutsche Mark

| German Credit | Original input (Bad credit risk) | Counterfactuals (Good credit risk) | | | | |
|---|---|---|---|---|---|---|
| Duration (months) | 24 - 72 | 4 - 12 | 12 - 24 | 24 - 72 | 12 - 24 | 12 - 24 |
| Credit amount (DM) | 3,410 - 18,425 | 1,570 - 3,410 | 1,570 - 3,410 | 1,570 - 3,410 | 3,410 - 18,425 | 1,570 - 3,410 |
| Age | 28 - 38 | 38 - 75 | 28 - 38 | 38 - 75 | 38 - 75 | 28 - 38 |
| Checking account (DM) | No account | 200+ | No account | 200+ | 200+ | 200+ |
| Credit history | Paid back duly | Paid back duly | Paid back duly | Paid back duly | Paid back duly | Paid back duly |
| *Purpose* | Retraining | Retraining | Retraining | Retraining | Retraining | Retraining |
| Savings account (DM) | Under 100 | 1,000+ | Under 100 | 500 - 1,000 | 100 - 500 | Under 100 |
| Present employment | 7+ years | 4 - 7 years | Unemployed | 4 - 7 years | 7+ years | 1 - 4 years |
| Installment rate | Under 20 | Under 20 | 35+ | 25 - 25 | 20 - 25 | 35+ |
| *Personal status* | Female not single | Female not single | Female not single | Female not single | Female not single | Female not single |
| Other debtors | Co-applicant | Co-applicant | guarantor | Co-applicant | Co-applicant | None |
| Present residence | 7+ years | 7+ years | 7+ years | 7+ years | Under 1 year | Under 1 year |
| Property | None | Car or other | None | None | None | None |
| Other installments | None | bank | None | None | None | None |
| Housing | Rent | Rent | own | For free | Rent | For free |
| No. existing credits | 1 | 1 | 4 - 5 | 1 | 1 | 4 - 5 |
| Job | Skilled | Skilled | Unemployed | Unemployed | Skilled | Skilled |
| *No. people being liable* | 0 - 2 | 0 - 2 | 0 - 2 | 0 - 2 | 0 - 2 | 0 - 2 |
| Telephone | No | No | Yes | No | Yes | Yes |
| *Foreign worker* | No | No | No | No | No | No |

Table 10: Counterfactual examples from Graduate Admission dataset.

| Graduate Admission | Original input (Low chance) | Counterfactuals (High chance) | | | | |
|---|---|---|---|---|---|---|
| GRE Score | 290 - 312 | 290 - 312 | 312 - 323 | 290 - 312 | 290 - 312 | 290 - 312 |
| TOEFL Score | 92 - 104 | 104 - 110 | 104 - 110 | 110 - 120 | 110 - 120 | 92 - 104 |
| Undergraduate GPA | 7 - 8 | 8 - 9 | 9 - 10 | 9 - 10 | 8 - 9 | 9 - 10 |
| *University Rating* | 2 / 5 | 2 / 5 | 2 / 5 | 2 / 5 | 2 / 5 | 2 / 5 |
| Statement of Purpose | 2 / 5 | 2 / 5 | 5 / 5 | 2 / 5 | 2 / 5 | 2 / 5 |
| Letter of Recommendation | 2.5 / 5 | 4 / 5 | 2.5 / 5 | 2.5 / 5 | 2.5 / 5 | 2 / 5 |
| Research Experience | No | No | No | Yes | No | Yes |

Table 11: Counterfactual examples from Student Performance dataset.

| **Student Performance** | Original input (Fail) | Counterfactuals (Pass) | | | | |
|---|---|---|---|---|---|---|
| Age | 16 - 17 | 16 - 17 | 16 - 17 | 16 - 17 | 16 - 17 | 15 - 16 |
| School absences | 0 - 4 | 0 - 4 | 0 - 4 | 4 - 32 | 0 - 4 | 4 - 32 |
| *First period grade* | 10 - 13 | 10 - 13 | 10 - 13 | 10 - 13 | 10 - 13 | 10 - 13 |
| Second period grade | 0 - 10 | 13 - 19 | 13 - 19 | 13 - 19 | 13 - 19 | 13 - 19 |
| *Mother's education* | Primary | Primary | Primary | Primary | Primary | Primary |
| *Father's education* | None | None | None | None | None | None |
| Weekly study time | 2 - 5 hours | 2 - 5 hours | 2 - 5 hours | 2 - 5 hours | 5 - 10 hours | 2 - 5 hours |
| *Family educational support* | No | No | No | No | No | No |
| Wanting higher education | Yes | Yes | No | No | Yes | Yes |
| Internet access at home | Yes | No | Yes | Yes | No | Yes |
| In a romantic relationship | Yes | No | No | No | Yes | Yes |
| Free time | Moderate | High | Moderate | Moderate | Moderate | Low |
| Going out frequency | Very often | Very often | Very often | Rarely | Very often | Often |
| Health status | Good | Good | Good | Very good | Good | Good |

Table 12: Counterfactual examples from SBA dataset.

| **SBA** | Original input (Low default risk) | Counterfactuals (High default risk) | | | | |
|---|---|---|---|---|---|---|
| Loan term in months | 60 - 84 | 60 - 84 | 60 - 84 | 60 - 84 | 84 - 240 | 60 - 84 |
| No. employees | 3 - 8 | 3 - 8 | 3 - 8 | 3 - 8 | 8 - 650 | 8 - 650 |
| No. jobs created | 0 - 2 | 0 - 2 | 2 - 130 | 0 - 2 | 2 - 130 | 0 - 2 |
| No. jobs retained | 0 - 1 | 4 - 535 | 0 - 1 | 1 - 4 | 0 - 1 | 4 - 535 |
| Charged off amount | 0 - 15K | 0 - 15K | Above 15K | Above 15K | Above 15K | 0 - 15K |
| Gross amount approved | 61K - 300K | 61K - 300K | 4K - 30K | 30K - 61K | 61K - 300K | 30K - 61K |
| Approved guaranteed amount | 42K - 240K | 42K - 240K | 42K - 240K | 42K - 240K | 15K - 42K | 42K - 240K |
| % gross amount guaranteed | 75 - 100 | 75 - 100 | 75 - 100 | 50 - 75 | 50 - 75 | 75 - 100 |
| *From Urban or Rural* | Urban | Urban | Urban | Urban | Urban | Urban |
| *New or Existing business* | No | No | No | No | No | No |
| Loan backed by real estate | No | No | No | No | No | No |
| *Active loan in recession* | No | No | No | No | No | No |

