# OpenReview forum: "Learning to Counter: Stochastic Feature-based Learning for Diverse Counterfactual Explanations"
_ICLR.cc/2023/Conference — Submitted to ICLR 2023_

### Official Review · Reviewer_Xqce · 2022-10-20

**Confidence:** 5
**Correctness:** 3
**Technical Novelty And Significance:** 2
**Empirical Novelty And Significance:** 2
**Recommendation:** 5

**Clarity, Quality, Novelty And Reproducibility:**

Clarity
=====
* The submission is clear and easy to understand. It lacks some implementation details but that could be easily fixed.

Quality
=====
* The overall quality is good.

Novelty
======
* Research-wise, this work does not introduce any new ideas (previous ideas are properly cited), this work is rather on the application side.

Reproducibility
===========
* The authors provide the code but the reproducibility could improve by providing more implementation details in the paper.



**Strength And Weaknesses:**

Strengths
=======
* The proposed method is sound
* The paper is clear and very well written, I really liked Table 1
* The computational cost is small
* I liked the OOD datasets experiment
* The authors provide the code

Weaknesses
=========
* The diversity measure is ill-defined (here and in previous state of the art). For instance, imagine an instance defined by N attributes and for which changing attribute number 1 generates a counterfactual. Now we can generate an arbitrary number of counterfactuals by modifying attribute number 1 and any other attribute (to increase hamming distance). However all these counterfactuals provide the same information (that the classifier is sensitive to attribute 1). I suggest you add some comment about this in the paper.
* From the paper, it is difficult to obtain all the details to reproduce their method. For instance, how is the selector instantiated? Is it just a sigmoid on top of a neural network? How is that neural network defined? Could you add implementation details (in general)?
* Research-wise, this work does not introduce any new ideas (previous ideas are properly cited), this work is rather on the application side.

**Summary Of The Paper:**

The authors propose a method to generate counterfactual explanations that are *valid, actionable, sparse, diverse, plausible,* and computationally *scalable*. They achieve this by learning an amortized local counterfactual distribution conditioned on the original data point. To learn it, they train a categorical counterfactual generator from which they sample the attributes to be modified with a selector that models a Bernoulli distribution. To enforce sparsity, they regularize the L1 norm of the selector. The proposed method achieves high validity and diversity rates compared with previous state of the art while having a relatively low computational cost.

**Summary Of The Review:**

The method is sound and the text is clear. My main concern is whether the author's contribution is significant enough for ICLR, I would appreciate if the authors could comment on that. The submission could also be improved by adding implementation details and a discussion about the limitations of this work. Thus, I temporarily recommend a score of 5 until these points are discussed.

---

> ### Author Response · Authors · 2022-11-16
> **Novelty of our L2C framework**
>
> We thank the reviewer for the time and valuable feedbacks. We here would like to explain the establishment of our novelty as follows:
>
> We propose a probabilistic learning-based framework for counterfactual generations. We model individual distributions for each feature so that when combined they form a valid counterfactual distributions, and adopt amortized inference to leverage efficiency in producing post-hoc explanations. Our work is different from existing works in the following ways:
>
> **1. Compared with  F-VAE (Mahajan et al., 2019); C-CHVAE (Pawelczyk et al., 2020); CRUDS (Downs et al., 2020); FAST-AR Verma et al., 2022):**
>
> Though sharing the technique of amortized inference, the underlying framework of each method is different.  F-VAE, C-CHVAE and CRUDS use VAE or VAE-based counterparts to model generative distribution while FAST-AR reformulates the problem into a Markov Decision Process. Meanwhile, our L2C directly optimizes for feature distributions using Gumbel reparameterization trick. To the best of our knowledge, this is the first time this strategy is used across related works and we are the first to tackle diverse counterfactual generations in an amortized optimization framework.
>
> **2.Compared with DICE (Mothilal et al., 2020) and COPA (Bui et al., 2022):**
>
> DICE and COPA are two of very few methods that tackle Diversity rigorously. However, as discussed, both methods are highly inefficient and does not consider the trade-off between Sparsity and Diversity. Our experiments extensively show that COPA is sub-optimal for all tasks and L2C is significantly more efficient than DICE while balances this trade-off more effectively.
>
> **3. Dealing with heterogeneous tabular data:**
>
> Given the rapid growth of works in this research area, being able to deal with heterogeneous tabular data is the minimum requirement a method should satisfy in order to be useful for practical applications. To the best of our understanding, Pawelczyk et al employ various likelihood model for suitable data types e.g., Gaussian for numerical features, Categorical for categorical features or Bernoulli for binary data. In our work, we propose to discretize the data for 3 main reasons:
>
> * We can conveniently apply one functional form feature-wise.
>
> * We believe it yields more useful explanations and easier for human users to follow the suggestions than forcing
> them to meet hard requirements from specific numerical values.
>
> * It helps expand the original input space that may later support better generalization.
>
> While Categorical sampling is natural to deal with categorical features, note that our purpose of Bernoulli sampling is different, which is discussed in the next section.
>
> **4. Bernoulli sampling to enforce Sparsity**
>
> To enforce sparsity in a generative counterfactual, prior works use standard metrics such as $\mathrm{L1}$ or $\mathrm{L2}$ to penalize the distance between the counterfactual and original data point. Verma et al. (2020) criticizes this approach as non-obvious, especially for handling categorical features. Avoiding the use of distance measures, we employ the Selector function to output the probability $\pi$ where each element $\pi_i$ represents the probability that the feature $i$ is mutated. We can then directly enforce sparsity by minimizing the regularized term $||\pi||_1$.
>
> **Finally**, we have conducted experiments on various real-world datasets demonstrating that we resolve the trade-off among counterfactual constraints more effectively than the baselines, specifically the capability of generating 100\% valid and diverse counterfactuals in a highly efficient manner.
>
> We however are very grateful to the reviewer for pointing this out. Motivated by your comments, we have improved our writing in related sections to give readers a clearer overview of how our method is different related works.

---

> > ### Author Response · Authors · 2022-11-16
> > **Other considerations**
> >
> > **The diversity measure is ill-defined (here and in previous state of the art). For instance, imagine an instance defined by N attributes and for which changing attribute number 1 generates a counterfactual. Now we can generate an arbitrary number of counterfactuals by modifying attribute number 1 and any other attribute (to increase hamming distance). However all these counterfactuals provide the same information (that the classifier is sensitive to attribute 1). I suggest you add some comment about this in the paper.**
> >
> > The whole idea of diversity is to explore different combinations of features and their values that can help counter the original prediction. Not only does it consider whether a feature gets changed but also what value the feature changes to. Our Diversity metric thus attempts to quantify this capability.
> >
> > *We would like to clarify that the goal of enforcing diversity is to provide a variety of suggestions to end-users so that they can select the best fit to their situations*. The nature of our problem is combinatorial and it is not trivial since not all combinations of value can change the original label. Every combination of features and their values provides a different improvement strategy for the user to achieve the expected outcome.
> >
> > **From the paper, it is difficult to obtain all the details to reproduce their method. For instance, how is the selector instantiated? Is it just a sigmoid on top of a neural network? How is that neural network defined? Could you add implementation details (in general)?**
> >
> > Let's first recall our problem setup. Given a data example $x$ with $N$ features, we discretize the numerical features into equal-sized buckets. We end up with an input of $N$ categorical features wherein each feature $x_i$ has $c_i$ levels. We apply one-hot encoding on each feature and flatten them into a single input vector $z \in \{0,1\}^D$ where $D = \sum^{N}_{i=1} c_i$.
> >
> > Let $\mathbb{K}$ denote the set of mutable features i.e., features that can be changed. We define the Selector $\mathcal{S}: \{0,1\}^D \mapsto [0, 1]^K$ mapping from the representation $z$ to the probability $\pi$ where $K = |\mathbb{K}|$ and  each element $\pi_i$ represents the probability the feature $i \in \mathbb{K}$ is mutable.
> >
> > We parametrize $\mathcal{S}$ with a $2$-layered neural network. Each layer consists of a $50$-dimensional dense layer, followed by a ReLU activation. The final layer takes a Sigmoid activation to produce a $K$-dimensional probability vector $\pi$.
> >
> > **Finally**, it is worth noting that this is the first time this idea is used across related works.

---

### Official Review · Reviewer_DTNp · 2022-10-20

**Confidence:** 4
**Correctness:** 3
**Technical Novelty And Significance:** 1
**Empirical Novelty And Significance:** 1
**Recommendation:** 3

**Clarity, Quality, Novelty And Reproducibility:**

The work is fairly well written, and the proposed method is clearly described. Unfortunately, the discussion of related work as well as the contextualization of the proposed solution with respect to related work are weak. The paper accumulates many ideas from previous works on counterfactual explanations. Various important aspects discussed in the paper are not novel and have already been proposed in previous papers.

Detailed feedback:
The improvements of the proposed framework over existing explanation baselines are not clear. The authors compare with a variety of methods that all have different goals (for example F-VAE and CRUDS versus FastAR). Therefore, I would suggest comparing with methods that primarily use (i) amortized inference to generate counterfactuals, and (ii) diversity constraints.

The only method that uses diversity constraints is DICE, but here the authors have not compared with the method suggested by Mothilal et al., (2020), but instead with a method that uses random search to find counterfactuals (that has been provided in the DICE library). To give the author’s a concrete idea of how a more meaningful evaluation could look like, I suggest considering the work by Pawelczyk et al (2021), who have split their evaluation according to the assumptions made by the underlying methods.

Finally, the work could be improved by focusing on fewer aspects and addressing them in innovative ways; instead, the proposed method addresses too many challenges at once by simply combining a wealth of already existing techniques into one greater system.


**Details Of Ethics Concerns:**

Since diversity of counterfactual explanations is an important aspect of the work, privacy-related risks should be discussed.

**Strength And Weaknesses:**

Strengths:
1. The addressed problem is relevant, timely and interesting to the XAI community.
2. The paper is fairly well written.
3. The methods’ evaluation is extensive.

Weaknesses:
1. The overview of related work as well as the contextualization of the proposed solution with respect to related work are weak.
2. The paper accumulates many ideas from previous works on counterfactual explanations; various aspects discussed in the paper are not novel and have already been proposed in previous papers:

(i) Amortized inference has been used in the following works:
Guo et al (2021),
Dan Ley et al (2022),
Verma et al (2022).

(ii) Dealing with heterogenous tabular data has been extensively studied in:
Karimi et al (2020)
Pawelczyk et al (2020) - Sampling categorical & Bernoulli data have also been used by Pawelczyk et al in their auto-encoder
Pawelczyk et al (2022) - Dealing with categorical variables.

(iii) Dealing with diverse counterfactuals was initially studied in Mothilal et al., (2020)

3. The discussion does not really critically address limitations or related improvement strategies. E.g.: What are possible extension strategies for handling domain constraints or expert knowledge? Do you see any tensions between diversity and privacy?...



**Summary Of The Paper:**

This paper presents a method to generate counterfactual explanations based on two components: (i) a feature selector and (ii) and an end-to-end network. While (i) makes sure that relevant features are selected, (ii) enables the model to enforce additional constraints (e.g., dealing with categorical variables).

**Summary Of The Review:**

Overall, the work lacks novelty and originality as well as critical discussions around limitations and shortcomings.

---

> ### Author Response · Authors · 2022-11-16
> **Novelty of our framework**
>
> We thank the reviewer for the detailed comments. We would like to explain the establishment of our novelty as follows:
>
> We here propose a probabilistic learning-based framework for counterfactual generations. We model individual distributions for each feature so that when combined they form a valid counterfactual distributions, and adopt amortized inference to leverage efficiency in producing post-hoc explanations. Our work is different from existing works in the following ways:
>
> **1. Compared with  F-VAE (Mahajan et al., 2019); C-CHVAE (Pawelczyk et al., 2020); CRUDS (Downs et al., 2020); FAST-AR Verma et al., 2022):**
>
> Though sharing the technique of amortized inference, the underlying framework of each method is different.  F-VAE, C-CHVAE and CRUDS use VAE or VAE-based counterparts to model generative distribution while FAST-AR reformulates the problem into a Markov Decision Process. Meanwhile, our L2C directly optimizes for feature distributions using Gumbel reparameterization trick. To the best of our knowledge, this is the first time this strategy is used across related works and we are the first to tackle diverse local counterfactual generations in an amortized optimization framework.
>
> **2.Compared with DICE (Mothilal et al., 2020) and COPA (Bui et al., 2022):**
>
> DICE and COPA are two of very few methods that tackle Diversity rigorously. However, as discussed, both methods are highly inefficient and does not consider the trade-off between Sparsity and Diversity. Our experiments extensively show that COPA is sub-optimal for all tasks and L2C is significantly more efficient than DICE while balances this trade-off more effectively.
>
> **3. Dealing with heterogeneous tabular data:**
>
> Given the rapid growth of works in this research area, being able to deal with heterogeneous tabular data is the minimum requirement a method should satisfy in order to be useful for practical applications. To the best of our understanding, Pawelczyk et al employ various likelihood model for suitable data types e.g., Gaussian for numerical features, Categorical for categorical features or Bernoulli for binary data. In our work, we propose to discretize the data for 3 main reasons:
>
> * We can conveniently apply one functional form feature-wise.
>
> * We believe it yields more useful explanations and easier for human users to follow the suggestions than forcing
> them to meet hard requirements from specific numerical values.
>
> * It helps expand the original input space that may later support better generalization.
>
> While Categorical sampling is natural to deal with categorical features, note that our purpose of Bernoulli sampling is different, which is discussed in the next section.
>
> **4. Bernoulli sampling to enforce Sparsity**
>
> To enforce sparsity in a generative counterfactual, prior works use standard metrics such as $\mathrm{L1}$ or $\mathrm{L2}$ to penalize the distance between the counterfactual and original data point. Verma et al. (2020) criticizes this approach as non-obvious, especially for handling categorical features. Avoiding the use of distance measures, we employ the Selector function to output the probability $\pi$ where each element $\pi_i$ represents the probability that the feature $i$ is mutated. We can then directly enforce sparsity by minimizing the regularized term $||\pi||_1$.
>
> **Finally**, we have conducted experiments on various real-world datasets demonstrating that we resolve the trade-off among counterfactual constraints more effectively than the baselines, specifically the capability of generating 100\% valid and diverse counterfactuals in a highly efficient manner.
>
> We however are very grateful to the reviewer for pointing this out. Motivated by your comments, we have improved our writing in related sections to give readers a clearer overview of how our method is different related works.

---

> > ### Author Response · Authors · 2022-11-16
> > **Other considerations**
> >
> > **The authors compare with a variety of methods that all have different goals (for example F-VAE and CRUDS versus FastAR). Therefore, I would suggest comparing with methods that primarily use (i) amortized inference to generate counterfactuals, and (ii) diversity constraints.**
> >
> > As discussed, none of the methods that leverage amortized inference has addressed the diversity constraint for local counterfactual explanations, which can only be found in non-amortized methods (e.g., DICE, COPA). We therefore decide to compare our method with both amortized baselines and non-amortized methods that address diversity to demonstrate that we have effectively resolved the trade-off between efficiency and diversity in a single novel framework.
> >
> > **The only method that uses diversity constraints is DICE, but here the authors have not compared with the method suggested by Mothilal et al., (2020), but instead with a method that uses random search to find counterfactuals (that has been provided in the DICE library).**
> >
> > As discussed in section 4.2, we report the performance of DiCE-Random because it is reported to consistently be the most efficient and highest-performing variant (Verma et al. 2022), compared to DiCE-Genetic and DiCE-KDTree. The same pattern is observed in our experiments. The following table reports Validity, Coverage and Time efficiency of DiCE-Genetic and DiCE-KDTree in comparison with DiCE-Random, which aligns with the reported results from Verma et al. (2022).
> >
> > |               | **Validity** | **Coverage** | **Time** |
> > |---------------|:------------:|:------------:|:--------:|
> > | **German**   |              |              |          |
> > | DiCE-Random   |    **100.00%**   |    **100.00%**   |   1150   |
> > | DiCE-Genetic  |    62.87%    |    90.24%    |   17615  |
> > | DiCE-KDTree   |     0.00%    |     0.00%    |   1010   |
> > |               |              |              |          |
> > | **Admission** |              |              |          |
> > | DiCE-Random   |    **100.00%**   |    **100.00%**   |    412   |
> > | DiCE-Genetic  |    92.91%    |    100.00%   |   3406   |
> > | DiCE-KDTree   |     0.00%    |     0.00%    |    420   |
> > |               |              |              |          |
> > | **Student**   |              |              |          |
> > | DiCE-Random   |    **100.00%**   |    **100.00%**   |   2518   |
> > | DiCE-Genetic  |    84.83%    |    100.00%   |   6171   |
> > | DiCE-KDTree   |     0.00%    |     0.00%    |   2493   |
> > |               |              |              |          |
> > | **SBA**       |              |              |          |
> > | DiCE-Random   |    **100.00%**   |    **100.00%**   |   5522   |
> > | DiCE-Genetic  |    72.42%    |    91.93%    |   7533   |
> > | DiCE-KDTree   |     0.00%    |     0.00%    |   5000   |
> >
> > **What are possible extension strategies for handling domain constraints or expert knowledge? Do you see any tensions between diversity and privacy?**
> >
> > Our framework does in fact take into account domain constraints on mutable and immutable features. As described in section 3.2 and Appendix A, we allow users to specify in advance which features cannot be changed and our method only modifies the features that are mutable while keeping immutable features at their original values.
> >
> > Regarding privacy issues, privacy is in fact an open challenge in this research area and to the best of our knowledge, has only begun to gain attention in recent years [1,2]. One of the main differences of our work is that we discretize data into categorical features. Thus, we can reduce the privacy risks of model stealing or sensitive data exposure when releasing the counterfactual suggestion to the public.
> >
> > We thank the reviewer for such an inspiring idea for future works. We have included the discussion on privacy and additional constraints in the revised paper.
> >
> > [1] Shokri, R., Strobel, M., & Zick, Y. (2021, July). On the privacy risks of model explanations. In Proceedings of the 2021 AAAI/ACM Conference on AI, Ethics, and Society (pp. 231-241).
> >
> > [2] Goethals, S., Sörensen, K., & Martens, D. (2022). The privacy issue of counterfactual explanations: explanation linkage attacks. arXiv preprint arXiv:2210.12051.

---

> > > ### Comment · Reviewer_DTNp · 2022-11-21
> > > **Thank you for your response.**
> > >
> > > Thank you for your response. After reading through your responses, I am still concerned about the novelty aspect, i.e., the work tries to tackle too many challenges by combining a plethora of existing techniques into a larger approach. I don’t think this is sufficient for a publication in a top-tier conference.

---

### Official Review · Reviewer_nrfX · 2022-10-25

**Confidence:** 3
**Correctness:** 3
**Technical Novelty And Significance:** 2
**Empirical Novelty And Significance:** 2
**Recommendation:** 5

**Clarity, Quality, Novelty And Reproducibility:**

The paper is well-written and easily follow in general. The idea is clearly stated and presented with experiments. The proposed method is interesting and, to some extent, novel. The code is provided for reproducibility.

**Strength And Weaknesses:**

Strength:
1) The proposed method addresses multiple constraints that counterfactual generation generally faces at the same time.
2) The optimization is amortized by two neural networks.
3) The evaluation metrics include various aspects of counterfactual generations. The results support the claim overall.
4) The discretization of continuous features looks interesting.

Weaknesses:
1) The metric, Manifold distance, is only measuring L1 distance among the nearest samples. I think one of the most important plausibility measurements is how much overlap the counterfactual sample lies in the target class in terms of distribution distance. L1 or L2 distance seems limited in regard to measuring plausibility.
2) The discretization of numerical features seems interesting to me. However, the trade-off of using discretization is unclear. In particular, how do we choose the level c_i? Does the choice of c_i affect the metrics? And are those comparisons evaluated in the discretized space or the original space?

**Summary Of The Paper:**

Counterfactual generation is a multi-model problem in the sense that satisfying all the constraints simultaneously is considered a challenging task. In this paper, the authors propose a stochastic feature-based learning approach to meet all the constraints. The proposed method discretizes each continuous feature and learns the generator and selector to generate diverse counterfactual samples. The generator learns a categorical distribution for feature-based perturbation. The selector serves to select feature-based selection distribution. They are multi-nomial and Bernoulli distributions, respectively. And they are both parameterized by a neural network. The loss is built based on the cross-entropy loss from the prediction and the target counterfactual class and an L1 norm that encourages sparsity. The proposed method are  compared with the most popular methods on 4 popular tabular data sets.

**Summary Of The Review:**

The paper proposes a novel method that can generate diverse counterfactual samples for understanding a pre-trained black-box classifier. The proposed method introduces two neural networks, a generator and a selector, that take the discretized input space. Although the discretization of the input space looks novel to me, the novelty is limited. The loss function that trains the generator and selector takes cross-entropy loss between the predicted label and target class label, and the L1 norm, which is not new at all.

---

> ### Author Response · Authors · 2022-11-16
> **We thank the reviewer for the useful comments.**
>
> We address the reviewer's questions in the following:
>
> **The metric, Manifold distance, is only measuring L1 distance among the nearest samples. L1 or L2 distance seems limited in regard to measuring plausibility.**
>
> Our current framework is concerned with finding local counterfactual examples. We therefore believe L1 or L2 is more relevant to measure the distance between a generated example with the corresponding input in the local space than measuring the global distance between two distributions.
>
> **However, the trade-off of using discretization is unclear. In particular, how do we choose the level c_i? Does the choice of c_i affect the metrics? And are those comparisons evaluated in the discretized space or the original space?**
>
> We first would like to clarify that for numerical features, comparisons are evaluated in the original continuous space to assure consistency with the baselines.
>
> In our work, numerical features are quantized such that each bucket contains at least $100$ data points to be well represented. Choosing very few is likely to cause under-fitting since there are very few useful combinatorial patterns that can counter the original label. We need sufficient diversity for effective learning. However, too many buckets are undesirable since it can hurt generalization due to some following reasons : (1) each bucket would contain too little data and the chosen middle value (used to decode one-hot vector as described in section 3.3) may not represent the bucket well, and (2) the model has more combinations of features to explore, thus can converge to sub-optimal combinations that cannot generalize well on unseen test points. The impact of discretization also depends on the proportion of numerical features in the data compared with categorical features. It would be less severe if there are more categorical features to provide sufficiently useful patterns for generalization.
>
> In this regard, we therefore decide to split data into equal-sized buckets in the hope of balancing the trade-off. We now demonstrate that this strategy yields stable results.
>
> We experimented with $3$ random bucket quantization variants: for each feature, we randomly select the number of buckets in the range $[2,10]$ and still select the variant wherein each bucket contains roughly at least $100$ data points. We report the results on German and Admission datasets in the following table, which shows there are not any significant fluctuations in the performance compared to the results under our reported settings. Furthermore, our method can still obtain high performances in these variants for all metrics. Note that here we do not set any sparsity threshold $\theta$ on the generated samples.
>
> |                  | **Prox Num** | **Prox Cat** | **Diversity** | **Sparsity** | **Validity** | **Coverage** | **Manifold** | **Valid Cat** | **Time (s)** |
> |------------------|:------------:|:------------:|:-------------:|:------------:|:------------:|:------------:|:------------:|:-------------:|:------------:|
> | **German**       |              |              |               |              |              |              |              |               |              |
> | Reported Setting |     0.806    |    30.46%    |     40.61%    |    32.90%    |    100.00%   |    100.00%   | 0.532        |    100.00%    |      13      |
> | Variant 1        |     0.890    |    32.25%    |     42.14%    |    35.06%    |    100.00%   |    100.00%   | 0.489        |    100.00%    |      12      |
> | Variant 2        |     0.857    |    33.30%    |     43.86%    |    37.31%    |    100.00%   |    100.00%   | 0.546        |    100.00%    |      13      |
> | Variant 3        |     0.849    |    32.87%    |     41.18%    |    37.11%    |    100.00%   |    100.00%   | 0.491        |    100.00%    |      11      |
> |                  |              |              |               |              |              |              |              |               |              |
> | **Admission**    |              |              |               |              |              |              |              |               |              |
> | Reported Setting |     1.259    |    31.95%    |     41.59%    |    48.12%    |    100.00%   |    100.00%   | 0.898        |    100.00%    |       5      |
> | Variant 1        |     1.252    |    33.32%    |     43.20%    |    49.17%    |    100.00%   |    100.00%   | 0.927        |    100.00%    |       4      |
> | Variant 2        |     1.229    |    33.60%    |     44.74%    |    49.45%    |    100.00%   |    100.00%   | 0.938        |    100.00%    |       5      |
> | Variant 3        |     1.242    |    32.77%    |     42.12%    |    48.82%    |    100.00%   |    100.00%   | 0.939        |    100.00%    |       5      |

---

### Official Review · Reviewer_67mR · 2022-11-04

**Confidence:** 4
**Correctness:** 2
**Technical Novelty And Significance:** 2
**Empirical Novelty And Significance:** 2
**Recommendation:** 3

**Clarity, Quality, Novelty And Reproducibility:**

Clarity: The paper is mostly easy to follow.

Quality: As mentioned in the previous section, usage of softmax to handle one-hot encoded features is a neat trick. However, the paper does not provide justification into why discretizing all features is a good idea and how the performance would be with high dimensional datasets.

Novelty: The Gumbel-softmax trick is well known but to the best of my knowledge, this is the first time this trick is used for generating CFs.

Reproducibility: The paper provides implementation details (e.g., model architecture) in appendix, and also provides a link to the code repo.

**Strength And Weaknesses:**

**Strengths:** Usage of softmax to ensure validity of one hot encoded features is a neat idea.

**Weaknesses:** Overall, the proposed methodology does not seem to solve the issues with plausibility. I also think some choices seem quite adhoc and need better justification. Please see detailed comments below:

1. As far as I could see, the distribution of each categorical feature is being learnt independently which may lead to unrealistic combinations (for instance, very young people with decades of work experience). The paper does not discuss how this issue should be resolved.

2. It is also not clear if discretizing numerical features is a very good idea. How would this scale to high dimensional datasets? How should the number of discretization buckets be chosen?

3. On a related note, discretized numerical features are converted back to the original representation by assigning them the middle value. Does this lead to any accuracy issues (since the precise feature value is lost)?

4. The usage of Bernoulli is an interesting idea but it also adds a few more knobs (every feature $i$ has a corresponding $\pi_i$). How could one handle sparsity at the scale of all the features?

5. The experimental setup makes a few assumptions that seem a bit questionable. I am not sure why selecting a random perturbation method (DiCE Random) is a good strategy. Similarly, assigning only 3 seconds for a single CF generation seems a bit unfair to posthoc CF generation methods. It is also not very clear to me why numerical and categorical plausibility is considered separately in Table 1 when there might be dependencies between them.

**Summary Of The Paper:**

The goal of the paper is to generate counterfactual explanation (CFs) of classification models. The paper points our several problems with existing CF generation methods, many of which are rooted in the fact that these methods are not able to handle categorical features particularly well. The previous methods also work in a posthoc manner which affects their runtime performance. The paper aims to solve these problems by training the CF generator. The crucial design choice is to discretize all feature as categoricals and learn their distributions. Usage of softmax ensures that the post-processing of one-hot encoded features does not lead to invalid CFs. The paper also shows experiments with several datasets showing better performance than existing methods.

**Summary Of The Review:**

Overall, the paper uses intriguing ideas to get around various issues for generating CFs. However, many choices seem adhoc and could be better justified. It is also not clear how well the setup would work with high dimensional datasets where discretization might lead to very large number of features.

---

> ### Author Response · Authors · 2022-11-16
> **Discretization and Feature Learning (1/2)**
>
> We thank the reviewer for the valuable comments. We address the concerns from the reviewer as follows:
>
> **The distribution of each categorical feature is being learnt independently which may lead to unrealistic combinations (for instance, very young people with decades of work experience).**
>
> Handling correlational or even causal constraints among features is indeed a non-trivial challenge, which often requires domain knowledge about the true structural causal models. To the best of our knowledge, there are only a few works addressing this, notably [1,2,3], by assuming access to partial knowledge or utilizing constraints from user specifications. However, there have been no proper quantitative metrics to evaluate whether the constraint has been satisfied effectively.
>
> As discussed in section 6, dealing with feature-wise causal relations is beyond the scope of our current work and our future works will focus on tackling extensions. In this paper, we attempt to demonstrate feasibility of our novel learning-based framework, through which future extensions can be done straightforwardly and more conveniently.
>
> For example, recall that in this work, we assume the **conditional distributions** of each feature $P(x_i|x)$ to be independent. We thus can model the joint distribution among any $k$ features as $P(x_1, x_2, ..., x_k |x) = \prod_{i=1}^k P(x_i|x)$. One can look to imposing constraints on this joint distribution among the related features to ensure the samples respect causal constraints.
>
> [1] Divyat Mahajan, Chenhao Tan, and Amit Sharma. Preserving causal constraints in counterfactual explanations for machine learning classifiers. arXiv preprint arXiv:1912.03277, 2019.
>
> [2] Karimi, A. H., Schölkopf, B., & Valera, I. (2021, March). Algorithmic recourse: from counterfactual explanations to interventions. In Proceedings of the 2021 ACM conference on fairness, accountability, and transparency (pp. 353-362).
>
> [3] Sahil Verma, Keegan Hines, and John P Dickerson. Amortized generation of sequential algorithmic recourses for black-box models. In Proceedings of the AAAI Conference on Artificial Intelligence,
> volume 36, pp. 8512–8519, 2022.
>
> **It is also not clear if discretizing numerical features is a very good idea. How would this scale to high dimensional datasets?**
>
> Tackling high dimensional datasets is in fact a common challenge for all methods in this line of work, and we believe it is more serious for methods that adopt greedy algorithms (e.g., DICE-Random) or linear integer programming (e.g., MACE). This in fact highlights the merit of amortized optimization frameworks like ours. Notice that the learning process for each feature is independent, thus in the case of high dimensional datasets, one can utilize parallel processing to improve efficiency. In this work, we have conducted extensive experiments on various common real-world datasets, which demonstrated the effectiveness of our methods for practical applications.
>
> **On a related note, discretized numerical features are converted back to the original representation by assigning them the middle value. Does this lead to any accuracy issues (since the precise feature value is lost)?**
>
> Given an input $x$, we associate the prediction on its generated counterfactual example $\tilde{x}$, which contains discretized features, with that on its continuous representation obtained through taking the middle values. During training, the middle values are directly evaluated against the black-box classifier, which assures accuracy consistency. Note further that the prediction on the $x$ is still obtained from its original representation, so there is no feature value lost here.

---

> > ### Author Response · Authors · 2022-11-16
> > **Discretization and Feature Learning (2/2)**
> >
> > **How should the number of discretization buckets be chosen?**
> >
> > In our work, numerical features are quantized such that each bucket contains at least $100$ data points to be well represented. Choosing very few is likely to cause under-fitting since there are very few useful combinatorial patterns that can counter the original label. We need sufficient diversity for effective learning. However, too many buckets are undesirable since it can hurt generalization due to some following reasons : (1) each bucket would contain too little data and the chosen middle value (used to decode one-hot vector as described in section 3.3) may not represent the bucket well, and (2) the model has more combinations of features to explore, thus can converge to sub-optimal combinations that cannot generalize well on unseen test points. The impact of discretization also depends on the proportion of numerical features in the data compared with categorical features. It would be less severe if there are more categorical features to provide sufficiently useful patterns for generalization.
> >
> > In this regard, we therefore decide to split data into equal-sized buckets in the hope of balancing the trade-off. We now demonstrate that this strategy yields stable results.
> >
> > We experimented with $3$ random bucket quantization variants: for each feature, we randomly select the number of buckets in the range $[2,10]$ and still select the variant wherein each bucket contains roughly at least $100$ data points. We report the results on German and Admission datasets in the following table, which shows there are not any significant fluctuations in the performance compared to the results under our reported settings. Furthermore, our method can still obtain high performances in these variants for all metrics. Note that here we do not set any sparsity threshold $\theta$ on the generated samples.
> >
> >
> > |                  | **Prox Num** | **Prox Cat** | **Diversity** | **Sparsity** | **Validity** | **Coverage** | **Manifold** | **Valid Cat** | **Time (s)** |
> > |------------------|:------------:|:------------:|:-------------:|:------------:|:------------:|:------------:|:------------:|:-------------:|:------------:|
> > | **German**       |              |              |               |              |              |              |              |               |              |
> > | Reported Setting |     0.806    |    30.46%    |     40.61%    |    32.90%    |    100.00%   |    100.00%   | 0.532        |    100.00%    |      13      |
> > | Variant 1        |     0.890    |    32.25%    |     42.14%    |    35.06%    |    100.00%   |    100.00%   | 0.489        |    100.00%    |      12      |
> > | Variant 2        |     0.857    |    33.30%    |     43.86%    |    37.31%    |    100.00%   |    100.00%   | 0.546        |    100.00%    |      13      |
> > | Variant 3        |     0.849    |    32.87%    |     41.18%    |    37.11%    |    100.00%   |    100.00%   | 0.491        |    100.00%    |      11      |
> > |                  |              |              |               |              |              |              |              |               |              |
> > | **Admission**    |              |              |               |              |              |              |              |               |              |
> > | Reported Setting |     1.259    |    31.95%    |     41.59%    |    48.12%    |    100.00%   |    100.00%   | 0.898        |    100.00%    |       5      |
> > | Variant 1        |     1.252    |    33.32%    |     43.20%    |    49.17%    |    100.00%   |    100.00%   | 0.927        |    100.00%    |       4      |
> > | Variant 2        |     1.229    |    33.60%    |     44.74%    |    49.45%    |    100.00%   |    100.00%   | 0.938        |    100.00%    |       5      |
> > | Variant 3        |     1.242    |    32.77%    |     42.12%    |    48.82%    |    100.00%   |    100.00%   | 0.939        |    100.00%    |       5      |
> >
> > **How could one handle sparsity at the scale of all the features?**
> >
> > Elaborating on the above discussion, we can in fact control sparsity by choosing a proper number of buckets. At a high level, a feature with a sufficiently large discretized buckets provides more useful combinations to counter the original prediction, making it more likely to be considered a mutable feature i.e., $\pi_i$ is closer to $1$. Note further that we can control sparsity of $\pi$ through the loss coefficient $\alpha$ on the second term.

---

> > > ### Author Response · Authors · 2022-11-16
> > > **Baseline Experiments**
> > >
> > > **I am not sure why selecting a random perturbation method (DiCE Random) is a good strategy.**
> > >
> > > As discussed in section 4.2, DiCE-Random is reported to consistently be the most efficient and highest-performing variant (Verma et al. 2022), compared to DiCE-Genetic and DiCE-KDTree. The same pattern is observed in our experiments. In the following table, we report Validity, Coverage and Time efficiency of DiCE-Genetic and DiCE-KDTree in comparison with DiCE-Random across datasets. They align with the reported results from Verma et al. (2022)
> > >
> > > |               | **Validity** | **Coverage** | **Time** |
> > > |---------------|:------------:|:------------:|:--------:|
> > > | **German**   |              |              |          |
> > > | DiCE-Random   |    **100.00%**   |   **100.00%**   |   1150   |
> > > | DiCE-Genetic  |    62.87%    |    90.24%    |   17615  |
> > > | DiCE-KDTree   |     0.00%    |     0.00%    |   1010   |
> > > |               |              |              |          |
> > > | **Admission** |              |              |          |
> > > | DiCE-Random   |     **100.00%**   |   **100.00%**   |  412   |
> > > | DiCE-Genetic  |    92.91%    |    100.00%   |   3406   |
> > > | DiCE-KDTree   |     0.00%    |     0.00%    |    420   |
> > > |               |              |              |          |
> > > | **Student**   |              |              |          |
> > > | DiCE-Random   |     **100.00%**   |   **100.00%**   | 2518   |
> > > | DiCE-Genetic  |    84.83%    |    100.00%   |   6171   |
> > > | DiCE-KDTree   |     0.00%    |     0.00%    |   2493   |
> > > |               |              |              |          |
> > > | **SBA**       |              |              |          |
> > > | DiCE-Random   |    **100.00%**   |   **100.00%**   |   5522   |
> > > | DiCE-Genetic  |    72.42%    |    91.93%    |   7533   |
> > > | DiCE-KDTree   |     0.00%    |     0.00%    |   5000   |
> > >
> > > **Assigning only 3 seconds for a single CF generation seems a bit unfair to posthoc CF generation methods.**
> > >
> > >
> > > It is worth noting that our time budget is set globally in order to compare the computational costs of methods in the fairest manner. It is in fact a practical constraint a method should address for it to be useful for real-world applications. Notice from Tables 2 and 3 in the main paper that not only our method L2C but DiCE and F-VAE can also generate multiple valid counterfactuals within the time limit.

---

### Decision · Program_Chairs · 2023-01-20

**Decision:**

Reject

**Justification For Why Not Higher Score:**

There are problems that come from the reviewer, as mentioned above, but there are also fundamental issues with the idea making sense at all. The structural basis of counterfactuals seems to be missed by the authors, as I mentioned above, which makes the paper hard to support.

**Justification For Why Not Lower Score:**

N/A

**Metareview: Summary, Strengths And Weaknesses:**

This paper presents a method to generate counterfactual explanations based on a feature selector and an end-to-end network.  The problem of having proper explanations in AI is one of the major challenges in the current AI literature, and the idea of using counterfactuals seems suitable for this purpose.

Still, the paper contains some problems. First, it put together different ideas from previous works on counterfactual explanations, while most aspects discussed are not novel and have already been proposed in previous papers. Second, some reviewers have problems with the discretization procedure and its justification. Third, the idea of having counterfactual explanations without causal content, as the authors seem to suggest, seems implausible. In other words, there is a conceptual gap between what the paper does and what is expected from explanations in the real world, which makes it hard to support such a work. One suggestion to improve this aspect is to study causality and how counterfactuals are derived from causal relationships, following, for example, Pearl’s work, and connect it with the proposal of this work.